# Whole Alga, Algal Extracts, and Compounds as Ingredients of Functional Foods: Composition and Action Mechanism Relationships in the Prevention and Treatment of Type-2 Diabetes Mellitus

**DOI:** 10.3390/ijms22083816

**Published:** 2021-04-07

**Authors:** Aránzazu Bocanegra, Adrián Macho-González, Alba Garcimartín, Juana Benedí, Francisco José Sánchez-Muniz

**Affiliations:** 1Pharmacology, Pharmacognosy and Botany Department, Pharmacy School, Complutense University of Madrid, 28040 Madrid, Spain; a.garcimartin@ucm.es (A.G.); jbenedi@ucm.es (J.B.); 2Nutrition and Food Science Department (Nutrition), Pharmacy School, Complutense University of Madrid, 28040 Madrid, Spain; amacho@ucm.es; 3AFUSAN Group, Sanitary Research Institute of the San Carlos Clinical Hospital (IdISSC), 28040 Madrid, Spain

**Keywords:** algae, diabetes, functional food, functional meat, microbiota, metabolism

## Abstract

Type-2 diabetes mellitus (T2DM) is a major systemic disease which involves impaired pancreatic function and currently affects half a billion people worldwide. Diet is considered the cornerstone to reduce incidence and prevalence of this disease. Algae contains fiber, polyphenols, ω-3 PUFAs, and bioactive molecules with potential antidiabetic activity. This review delves into the applications of algae and their components in T2DM, as well as to ascertain the mechanism involved (e.g., glucose absorption, lipids metabolism, antioxidant properties, etc.). PubMed, and Google Scholar databases were used. Papers in which whole alga, algal extracts, or their isolated compounds were studied in in vitro conditions, T2DM experimental models, and humans were selected and discussed. This review also focuses on meat matrices or protein concentrate-based products in which different types of alga were included, aimed to modulate carbohydrate digestion and absorption, blood glucose, gastrointestinal neurohormones secretion, glycosylation products, and insulin resistance. As microbiota dysbiosis in T2DM and metabolic alterations in different organs are related, the review also delves on the effects of several bioactive algal compounds on the colon/microbiota-liver-pancreas-brain axis. As the responses to therapeutic diets vary dramatically among individuals due to genetic components, it seems a priority to identify major gene polymorphisms affecting potential positive effects of algal compounds on T2DM treatment.

## 1. Introduction

Type-2 diabetes mellitus (T2DM) is a metabolic disorder characterized by an imbalance in blood glucose level and an altered lipid profile. The study of T2DM has become a priority, given its high prevalence (around 90% of diabetes mellitus cases in worldwide) and complexity [1]. Among the factors intervening in the development of T2DM, lifestyle, diet (high-fat, high sugar, and high-energy consumption), obesity, gut microbiota (GM), and genetics are important contributors of the pathology development [2,3,4,5].

Related to dietary habits, globalization has conditioned changes in food intake, with algae consumption, either as a food or as dietary supplement, being increasingly frequent in western countries [6,7,8]. Algae contain a variety of nutrients and phytochemicals that exhibit various biological activities [9,10]. Algae-rich diets traditionally eaten by Asians has consistently been related to a lower incidence of chronic diseases such as cancer, cardiovascular, and heart disease [11,12]. There is a very active line of research studying the modulation of nutrients and bioactive components on gene expression, a science known as nutrigenomics. In addition, the components of the diet can modulate epigenome producing permanent changes in gene expression, and metagenomics referring to the modulation of the microbiota [13,14,15]. Recent studies have shown that, besides genetic predisposition and diet, the GM affects glucose and lipid metabolisms as well as influences the balance between proinflammatory and anti-inflammatory effectors in the liver, affecting T2DM [16,17]. Given the richness of algae in bioactive compounds, it is a very promising component in these new areas of research.

Nowadays, there is a high interest in healthy natural products, including algae, which could prevent the appearance of T2DM and its comorbidities [18]. In fact, algae and their bioactive compounds are added as ingredients of functional foods (term coined for such foods with potential health benefits) [19,20]. A regular consumption of functional foods appears to be associated with improved antioxidant enzymes, suppress over production of proinflammatory cytokines, insulin sensitivity, and hypocholesterolemia functions, which are considered essential to preventing and controlling T2DM [20,21]. There have been indications that algae could be used as antidiabetic foods/ingredients, but the mechanisms of action remain unclear. Likewise, it is important to differentiate between including a whole alga or an extract, because the extraction process determines the subsequent activity observed, or their isolated compounds [22,23]. Despite their potential health benefits, recent controversy exists about the utility of algae as some adverse events have been associated with their consumption. Biochemical characteristics of the cell wall of algae, rich in polysaccharides and proteins with anionic carboxyl, sulfate, and phosphate groups, make them important bio sorbents of toxic heavy metals (As, Pb, Cd, etc.) from industrial wastewater. In addition, certain seaweeds can present dangerously high concentrations of iodine [24,25,26].

## 2. Type-2 Diabetes Mellitus

T2DM pathophysiology is due to a progressive loss of adequate β-cell insulin secretion frequently on the background of insulin resistance (IR) by tissues such as skeletal muscle, liver, and adipose tissues (Figure 1) [27]. T2DM is clinically diagnosed by the presence of hyperglycemia, increasing the risk of developing chronic complications, which are due to complex and interconnected mechanisms between hyperglycemia, IR, low-grade inflammation, and accelerated atherogenesis [28].

Metabolic imbalances linked with IR are based on the outcome of metabolic function of insulin on carbohydrate, fat, and protein. Under normal conditions, glucose uptake by peripheral tissues occurs through the InsR/IRS/PI3K/AKT pathway, whose defective activity is responsible for IR. Briefly, when insulin reaches the target cell, it binds to an insulin receptor (InsR), which autophosphorylates and induces insulin receptor substrate (IRS) activation. This conformational change allows the binding of phosphatidylinositol 3-kinase (PI3K) through its SH2 domain, whose activation promotes the PI3,4,5-trisphosphate (PIP3) formation in the plasma membrane. PIP3 serves as a binding site for AKT, a membrane-displaced protein that is phosphorylated at residues Ser^473^ and Thr^308^. AKT is a major contributor in insulin signaling, whose phosphorylation enables the translocation of the type 4 glucose transporter (GLUT 4) to the plasma membrane and regulates glycogen synthesis through glycogen synthase kinase-3 (GSK3) and glycogen synthase (GS) [29,30]. If the disease continues to progress, the β-cells degenerate and insulin production is reduced, leading to less glucose uptake at the tissue level and greater production and release of that substrate at the liver, reaching a vicious cycle that worsens the imbalance in glucose metabolism until it leads to T2DM.

Dyslipidemia is a cluster of serum lipids and lipoproteins alterations highly prevalent in T2DM patients and characteristics of metabolic syndrome (MS) [31,32]. This T2DM dyslipidemia cluster implies elevated levels of triglycerides (TG) as large very-low-density lipoproteins (VLDL1), low high density lipoprotein cholesterol (HDL-C) levels and raised amounts of dense and small low-density lipoproteins (LDL). Together, these alterations are known as “the atherogenic lipid triad” [33]. This lipid triad is associated with IR, central obesity, and non-alcoholic fatty liver disease (NAFLD) and relates to excessive flux of free fatty acids (FFA) from visceral adipose tissue that also contributes to IR and VLDL1 formation. VLDL1 particles are more exposed to the cholesterol ester transfer complex (CETP) that metabolically helps form TG rich in high-density lipoprotein (HDL) and small, dense LDL [34,35]. The increase of plasma TG drives the changes of core lipids between TG rich lipoproteins (TRLs) and HDL particles. There is a transfer increase of cholesterol ester and TG between HDL and TRLs by means of CETP, resulting in the triglycerides enrichment of the latter. HDL-TG are good substrates for hepatic lipase and the hydrolysis produces smaller HDL particles. The catabolic rate of the small HDL, is faster than that of normal HDL, resulting in a reduced amount of circulating HDL particles [33]. Differing from normal situations, IR reduces lipoprotein lipase (LPL) activity giving rise to prolonged VLDL half-life but increases hepatic lipase activity that enhances HDL enriched TGs catabolism originating from small and dense HDL particles, with little antioxidant power and a shorter half-life. The lipid changes associated with T2DM are attributed to increased FFA flux secondary to IR and aggravated by increased inflammatory adipokines [36].

High TG and low HDL-C levels with the TG/HDL-C ratio are parameters normally considered to diagnose T2DM, suggesting the importance of lipoprotein metabolism in this pathology [37,38]. In addition, Apo-B100 availability is increased in IR contributing to the small LDL particle. As these particles are depleted in cholesterol, determining LDL-C in T2DM has little diagnostic value [33].

T2DM development depends on various genetic, environmental, socioeconomic, and lifestyle factors [39]. Genetic charge plays a vital role in this disease, thus, individuals with diabetic first-degree relatives are four to six times more likely to develop T2DM. Although several genes participate in the physiopathology of T2DM, some gene variants play an outstanding role on T2DM, like the insulin receptor substrates (IRS1 and IRS2), those for the β-adrenergic receptor (ADRB2 and ADRB3), those for uncoupling proteins (UCP2 and UCP3), and those for the receptor activated by peroxisome proliferator-activated receptor alpha (PPAR-γ) [40]. Nonetheless, FTO, apoC3, ApoA4, and ApoA5 polymorphisms among others, appear related to the lipoprotein alterations observed in T2DM [41,42].

In relation to lifestyle, T2DM and its associated complications can be prevented or delayed through regular food intake and physical activity. In these sections the role of some nutrients and bioactive components should be discussed with special mention of algal components, which can be considered functional as they could improve glycemic control, activation of antioxidant enzymes, GM, and reduce over production of proinflammatory cytokines during T2DM [43].

### 2.1. Microbiota Dysbiosis

During the last decade, many studies have demonstrated that GM plays key role in the pathophysiology of T2DM. Considering that GM physiologically modulates carbohydrate metabolism, it is not strange to think that alterations in the microbiome can contribute to developing T2DM. In fact, both diabetic patients and animal models of T2DM have exhibited intestinal dysbiosis. Whether dysbiosis appears because of the pathology, or on the contrary, it acts as a risk factor and contributes to T2DM establishment has not been fully described [44,45]. Although it has not been resolved, there is sufficient evidence to think that certain changes in the microbiota’s composition participate in developing T2DM, which explains that dysbiosis is already presented in individuals with prediabetes [46]. Furthermore, in a prospective controlled trial, diabetic patients had postprandial endotoxemia years before the pathology onset comparing to non-diabetic ones, evidencing that losing colonic homeostasis occurs earlier [47].

There are numerous publications that analyze characteristic GM of T2DM, respect to the microbiome of healthy individuals. Among these studies, there is great variability, making it difficult to generalize. Most studies agree that the diversity of bacteria decreases in individuals with prediabetes or T2DM compared to healthy individuals [48]. Additionally, many studies report the increase in the Firmicutes/Bacteroidetes ratio in T2DM, like its increase in obesity and MS. However, the association of particular taxonomic groups with T2DM is difficult to establish [49]. Notably, the microbiome, beside health condition, is affected by genetic and environmental factors, age, race, geographic location, and lifestyle (diet and exercise), which function as confounding factors [50]. Finally, oral antidiabetic drugs influence the composition of GM very diversely, which contributes to variability and complicates the interpretation and drawing of general conclusions [51]. Taking this into account, Gurung et al. [49] have reviewed 42 human studies and established genders related to T2DM. According to these authors, the genera *Bifidobacterium*, *Bacteroides*, *Faecalibacterium*, *Akkermansia*, and *Roseburia* are reduced in patients with T2DM; whereas *Ruminococcus*, *Fusobacterium*, and *Blautia* are increased. Furthermore, the possibility of identifying specific negative genders associated or positively associated with the pathophysiology of T2DM may be key when designing therapeutic strategies with prebiotics and probiotics.

The evaluation of the altered functions linked to dysbiosis is as important as the characterization of the microbiome in patients with T2DM. Dysbiosis has been involved in the development and maintenance of IR, promoted by the dysregulation of key mechanisms. As indicated, GM actively participates in glucose homeostasis, highlighting its role in processes such as: modulation of incretin secretion, production of short-chain free fatty acids (SCFAs), bile acids metabolism and regulation of adipose tissue, liver, or skeletal muscle functions [51]. Changes in the intestinal ecosystem that lead to dysbiosis consequently origin the disturbance of these functions and the appearance of colonic permeability, colon, liver, and adipose tissue inflammation, alterations in insulin secretion, and IR, and globally disrupting glucose and lipid metabolism [49].

### 2.2. Genetics

Although T2DM is a chronic pathology of multifactorial origin, including environmental factors such as diet or physical activity, there is also a genetic implication that predisposes or causes said pathology [52]. Nowadays, it is estimated there is a heritability of 30–70%, thus it is important to know and identify the different genetic variants of the disease risk [53]. Advances in genetics, and especially in developing genome-wide association studies (GWAS) and sequencing, have made it possible to identify multiple risk variants at loci involved in T2DM development. In general, the GWAS have focused their efforts on finding common variants (minor allele frequency [MAF] > 5%); whereas developing direct sequencing of complete genomes or exomes elucidates protective alleles, and low (0.5% < MAF < 5%) and rare frequency risk (MAF < 0.5%) [54]. The different studies published have collected approximately over 300 loci associated with an increased risk of developing T2DM. Curiously, despite what is expected in the pathology where IR is the central basis, the vast majority of the identified loci are mainly associated with β-cell function and insulin secretion [52,53,55,56]. As summarized by Ingelsson et al. [55], the T2DM genetic architecture is characterized by several causal variants, most with only a small additive effect on risk. Therefore, in this review, the main gene variants that could most decisively affect this disease development will be summarized (Figure 2).

The T2DM multifactorial origin makes it difficult to categorize each of the genetic variants that increase the disease risk. A clear example is found in the association between IR and obesity, which is often inseparable [57]. Therefore, some authors have proposed to further delimit the T2DM-loci associated, relating them to specific functions such as fasting glucose, fasting insulin, glycated hemoglobin (HbA1c), β-cell functionality, or IR [52,58]. Dupuis et al. [58] performed a meta-analysis of 21 GWAS in up to 46,186 non-diabetic participants, to identify variants involved in glucose homeostasis and that, therefore, could increase T2DM risk. These authors found 12 independent loci associated with fasting glucose and/or β-cell functionality at genome-wide significance levels; G6PC2, MTNR1B, GCK, TCF7L2, SLC30A8, and ADCY5 were highlighted as the six loci with the major effect on fasting glucose. Likewise, they also reported that the alleles with the strongest effects on fasting glucose were associated with a higher T2DM risk (*p* < 5 × 10^−8^) [58].

The Genetics of Type-2 Diabetes (GoT2D) and Type-2 Diabetes Genetic Exploration by Next-generation sequencing in multi-Ethnic Samples (T2D-GENES) Consortia performed whole genome sequencing in 2657 Europeans (1326 with T2DM and 1331 as controls), and exome sequencing in 12,940 subjects [59]. These authors found 126 variants at four loci associated with T2DM, which included two common variant loci (TCF7L2, and ADCY5), a low-frequency variant in CCND2 (rs76895963), and a new common variant association close to EML4 [59]. Likewise, Mahajan et al. [60] studied 32 GWAS of the European population, in which they covered a population of 74,124 with T2DM and 824,006 controls. These authors found 152 loci with genome-wide significance (*p* < 5 × 10^−8^), excluding possible confounding factors such as body mass index. These analyzes revealed that TCF7L2 (rs7903146) could be because of the common variant with the most evidence and effect for T2DM in Europeans. In addition, they also identified 56 low-frequency and 24 rare T2D-associated variants across 60 loci. Of these 80 mapped signals, five belonged to variations widely described in the literature (TCF7L2, CCND2, INS-IGF2, KCNQ1, and CDKN1C), and two rare alleles not previously reported with large odd-ratios (KIF2B and DENND2C) that require further in vitro and in vivo validation [60].

This knowledge of the T2DM genetic variants opens new doors to pharmacogenomic and nutrigenomic strategies [61,62]. Thus, this review will focus on this second point, highlighting the precision nutrition importance, especially all those improvements after seaweed consumption.

## 3. Algae

### 3.1. Algae Definition and Classification

Algae are not considered a phylogenetic concept, but are a loose set of significant organisms, which have different origins, evolutionary lines, and biochemistry; also may have any or all of these characteristics making them groupable: simple, photosynthetic, aquatic vegetative structures without a vascular system, and reproductive bodies lacking a sterile layer of protective cells [63]. Like plants, they use energy from sunlight, they are present in both fresh and salt waters [63,64], both prokaryotes and eukaryotic taxa are included, and there is a wide range of vegetative morphologies varying in size. In addition, the divisions are distinguished from each other based on characteristics including photosynthetic pigments, starch like reserve products, cell covering, and other aspects of cellular organization [63,65] (Table 1).

Depending on its sizes, algae can be classified as unicellular or colonial microalgae, or multicellular marine organisms (macrophytes seaweeds).

Macroalgae are typically classified based on their chemical and morphological characteristics and mainly based on the specific pigments part of its composition. Thus, they are grouped into one of the three alga divisions: brown algae (also known as kelp. Phylum Ochrophyta, class Phaeophyceae), red algae (phylum Rhodophyta, class Rhodophyceae), and green algae (phylum Chlorophyta, classes Bryopsidophyceae, Chlorophyceae, Dasycladophyceae, Prasinophyceae, and Ulvophyceae) [66]. The brown or yellow-brown color of the brown algae is due to fucoxanthin; red algae often have brilliant color due to phycoerythrin and phycocyanin, which are dominant over the other pigments, chlorophyll a, β-carotene and several xanthophylls; green algae contain chlorophyll a and b and various characteristics xanthophylls [65,66].

However, classification of microalgae according to their pigments allows to distinguish between red microalgae (Rhodophyta) which most representatives belong to the genus *Porphyridium*. Green microalgae (Chlorophyta) such as Chlorella (genus *Chlorella*), and Cyanobacteria microalgae that are also called blue-green algae, may be considered seaweed, however, they evolved differently from macroalga. Among them, the microscopic forms of phylum Cyanophyta (Cyanophyceae) stand out, especially the genus *Arthospira* (Spirulina) and *Nostoc* [67].

### 3.2. Algal Consumption and Commercial Importance

Algae have been part of the human diet for thousands of years, based on archaeological evidence [68]. Seaweeds have long been traditionally used for food in certain Asian regions (traditionally in China, Japan, and the Republic of Korea) in soups or it used to wrap sushi (Nori)—a practice now spreading to other countries. It has also traditionally been consumed in European coastal communities (for example, in France, Norway, Wales, and Ireland) [69]. The demand for edible seaweed is increasing in community markets and new production models and new market trends are emerging [69]; further, it is found in restaurants and on supermarket shelves in many non-Asian countries. Consumers are incorporating recipes based on “seaweed” increasing its acceptance and popularity thanks to its high content of proteins, and minerals and because they are considered healthy and natural [18,69]. For example, the direct consumption of brown algae as a human food dates back years and continues today as a valuable food ingredient [70,71].

Due to the high nutritional and pharmaceutical values, seaweeds are traditionally consumed as food or as herbal medicines, along with many other uses; thus, about 221 species of seaweed are of commercial value. In China, algae cultivation is intensive because of its high consumption as food. Over 70 edible species have been reported in the Chinese diet, but only a selection of these are approved for food in the European Union or its Member States [18,65]. In addition, the current range of uses of algae has caused their growing cultivation, for example compounds derived from algae (in cosmetics and food) are on the rise [18,65]. In Europe, the most exploited algae species are *Laminaria hyperborea*, *Laminaria digitata*, and *Ascophyllum nodosum*. Spain, Portugal, and Germany are the major providers of algae fit for human consumption within Europe. With algae fit for human consumption, the main non-EU suppliers are Chile (approximately 2500 tons in 2015) and China (800 tons) [67,69]. Microalgae (*Spirulina* spp.) is also cultivated, although it is much under-reported. Australia, India, Israel, and Japan are among the producers of *Spirulina* [67].

The world production of aquatic plants reached 32.9 million tons in 2017. Farmed aquatic plants included mostly seaweeds (40 aquatic algae) and a much smaller production volume of microalgae (89.000 tons of farmed microalgae) [72]. About 31 million tons of seaweeds and other algae were harvested globally for direct consumption (for example: kelps, *Undaria pinnatifida* (Wakame), *Pyropia* spp. (note: some seaweed classified as genus *Porphyra* are now classified as genus *Pyropia*), and *Caulerpa* spp.) or further processing like raw material for the extraction of carrageenan (*Kappaphycus alvarezii* and *Eucheuma* spp.) [68,72]. In 2016, China and Indonesia stood out as major producers of aquatic plants. The most widely cultivated species include Japanese Kelp or Konbu (*Laminaria japonica*), Eucheuma seaweeds, Elkhorn sea moss (*Kappaphycus alvarezii*), *U. pinnatifida*, *Gracilaria* spp., and Nori (*Pyropia* species) all of them cultivated mainly for nutritional purposes [72]. With the cultivation of microalgae, *Spirulina* spp., *Chlorella* spp., *Haematococcus pluvialis*, and *Nannochloropsis* spp., are being marketed for production of human nutrition supplements and other uses [72].

The composition of seaweeds is highly variable, depending not only on the species but also on the time of its collection and their habitat [72]. These differences in the composition and concentration of the bioactive compounds present in the different species of seaweed seem responsible for the potential health benefits [11,73]. Algal bioactive compounds of commercial interest are not present in terrestrial food sources, include pigments, lipids, polyunsaturated fatty acids (PUFAs), different proteins (lectins, phycobiliproteins, peptides, and amino acids), polysaccharides, sterols, and polyphenols [11]. The biological activities reported for these components are varied [65,66]. Nonetheless, the presence, form, and level of bioactive compounds of natural algal populations are further influenced by many factors like temporal changes and reproductive development [74].

### 3.3. Algae as Functional Foods or as a Potential Raw Material for Bioactive Ingredients

Preventive and therapeutic approaches to T2DM focus on a holistic strategy that includes promoting a balanced diet combined with pharmacological intervention to reduce/control hyperglycemia, obesity, and cardiovascular complications [75]. Prevalence of obesity in developed countries causes greater research and design of new ways to treat pathologies involved such as T2DM, cardiovascular disease (CVD) and hypertension [76]. Therefore, the relationship between diet and health has become an important target that allows preventing, or even treating, some of these diseases. Functional foods or ingredients are one of the easiest ways to achieve reduced symptoms, even in T2DM [43]. Thus, several products have been reformulated with bioactive compounds or ingredients (e.g., algae have been incorporated in some food matrices, like meat products, to obtain functional meat), in which seaweeds provide a wide variety of positive health characteristics [77].

Due to their technological, organoleptic, and nutritional properties, algae are ideal to be added to many foods to increase their protein and other nutritional contents (salad dressings, beverages, and baked goods) and/or sold as protein supplements [68]. In fact, there are several studies that treat the application of algae as functional foods, with interesting conclusive data, in relation to associated comorbidities and CVD, but new studies presented expand their application in the treatment of T2DM, hence, the interest of a review of the latest data in relation to this pathology, their comorbidities, and algae. In addition, protector mechanisms of algae consumption against T2DM remains poorly understood, thus they will be also reviewed. Advance in understanding the role of specific foods as algae, their nutrients and bioactive compounds in the pathogenesis of T2DM is of supreme importance.

Evidence is documented on the critical role of the GM in regulating liver function and development of obese-related metabolic diseases, such as T2DM, IR, and hyperlipidemia. Therefore, the potential of therapies directed at the GM in these diseases is important. Among the different ways that exist to manipulate the intestinal microbiota, active components of algae are included [78]. However, to the best of our knowledge, no previous review has extended the scope and included its effect on GM. The potential of manipulating the GM in this metabolic disorder is assessed, with an examination of the latest and most relevant evidence relating to seaweeds.

Considering that an integrated approach including multiple biomarkers, genetic variability, effect of GM, and risk/benefit assessment should support the potential health effects of functional foods [79]. This review aims to study the potential preventive or therapeutic action of algae extracts or their isolated compounds against T2DM. In this narrative review, we summarize the evidence from the latest studies on evaluating the effect of the consumption of algae, its nutrients and non-nutrients on the control and management of T2DM, as well as its complications. We also discuss mechanisms and the possibility of using algae as functional foods or nutraceuticals. In addition, scientific experiments performed on seaweed used as functional food are presented.

## 4. Algal Composition, Structure, and Beneficial Effects on Type-2 Diabetes Mellitus

Diet quality and a healthy lifestyle are recommended for the prevention of most chronic degenerative diseases. As already mentioned, T2DM is a metabolic condition characterized by chronic hyperglycemia and results from the interplay of nutritional, environmental, and genetic factors [39]. The role of dietary energy, macronutrients, micronutrients, and bioactive compounds in developing T2DM have been widely studied [80,81]. Epidemiological studies have shown that western diets, rich in saturated fatty acids (SFA) and poor in fiber, are associated with increased risk of chronic diseases such as T2DM [82,83], whereas high intakes of carbohydrate and protein, especially animal protein, enhance the risk for T2DM [84].

This section review information on major dietary factors affecting T2DM and its potential relationship with algae composition. Algae are rich in polysaccharides, minerals, and vitamins. Their fatty acid profile consists mainly of ω-3 PUFAs and they are considered an excellent source of dietary fiber and protein [10,85]. Most have been studied in relation to the prevention or improvement of lipoprotein profiles and CVD risk and T2DM [10,86]. However, different factors can influence the health benefits of algal-derived food products. Crucial factors such as harvest, geographical regions, seasonal storage, and food processing techniques will influence the nutritional composition of algal species and their derivatives [68].

### 4.1. Energy

Excess energy consumption has been related to obesity and MS. Obesity is a critical risk factor of pre-diabetes and T2DM [87]. In obese individuals with android/central fat distribution, adipose tissue releases increased amounts of non-esterified (free) fatty acids (FFA), glycerol, hormones, proinflammatory cytokines, and other factors involved in IR development. Low caloric intake remains a crucial factor to improve insulin sensitivity and reduce body weight [88]. Furthermore, energy consumption is a main dietary factor in the development and complications of T2DM, because its excess consumption is related to obesity [87], and modifications in the lipoprotein profile [89]. Some suggest that IR is a key contributor to MS, a cluster of medical conditions in which overweight/obesity, dyslipidemia, hyperglycemia, and hypertension also play important roles and contribute significantly to CVD risk [57,89].

Seaweeds and microalgae are considered adequate food for obesity and T2DM patients [90]. Their high-water content (70–90%) [18], with their large amount of fiber [10], conditioning the low energy contribution of these non-terrestrial vegetables.

### 4.2. Polysaccharides and Fiber

Algae contain polysaccharides as the main component of biomass (33–47% of air–dried seaweeds), whose purpose is to store energy and function as structural elements, hence and because of its many health benefits, a great deal of attention has been paid to the isolation and characterization of polysaccharides from seaweed [10,12,68,91]. Despite the relatively high carbohydrate content in algae, starch is limited in alga whereas the dietary fiber content is high, ranging from 23 to 75% of dry weight, making them a valuable source of dietary fiber [92]. Nonetheless, as previously commented, dietary fiber in algae can be affected by multiple factors such as geographical origin, seasonal variations, time of harvesting, and by post-harvesting process [10].

Polysaccharides in algae differ from those of terrestrial plants. This makes the polysaccharide composition one of the fundamental characteristics used to classify algae (Table 1). Structural polysaccharides present in algal cell walls comprise water-insoluble high molecular-weight compounds (cellulose, xylan, and mannan) and water-soluble polysaccharides such as agar, carrageenan, alginate, fucoidan, pyropian, and ulvan, which represent most of the dietary fiber present in algae [92]. Soluble fractions comprise 52–56% of total fiber in commonly used green and red macroalgae, and 67–85% in brown macroalgae [68]. However, data discrepancies are found on the fiber content of algae, probably because of the extraction methods used. Still, concentrations of total and soluble fiber are higher in algae than those found in common vegetables [9]. Polysaccharides, as polymeric structures, are formed by repeating units of neutral and acidic sugars linked by specific glycosidic bonds [91]. In contrast to terrestrial plants, the polysaccharides of the marine algae are mainly sulfated. Remarkably, the glycosidic linkages, monosaccharide composition, molecular weight, and sulfate content can differ for the same polysaccharide depending on the algal species and harvesting time [93]. Nevertheless, valuable properties of polysaccharides usually depend on the sequence of monomer units in their molecules [91]. Algal polysaccharides present a low glycemic index (GI) [94]. In addition, seaweed decreased the GI and/or glycemic load of diets [94]; this factor with the high fiber content makes algae adequate food for T2DM treatment [95]. Thus, there is unmistakable evidence on the role of a low-GI diet related to the type of carbohydrates ingested in patients with T2DM [96]. Ojo et al. [96] conducted a systematic review and a meta-analysis showing that a low-GI diet is more effective in controlling HbA1c and fasting blood glucose than a high GI diet. Likewise, Zafar et al. [97] found that low GI diets were effective at reducing HbA1c, fasting glucose, body mass index, total cholesterol, and LDL-C, but did not affect on fasting insulin, HOMA-IR, HDL-C, TG, or insulin requirements. In addition, Ojo et al. [98] observed that a low-GI diet significantly decreased interleukin (IL) 6 in patients with T2DM compared to a high GI diet. A high-carbohydrate diet, rich in fiber and with a low GI/low glycemic load, may also be recommendable to treat T2DM [95].

The main polysaccharides obtained from marine algae are alginate, agar, and carrageenan, which they all are gelling algal polysaccharides. Due to their ability to form highly viscous solutions and gels, those hydrocolloids have a wide range of applications in food, pharmaceutical industries, and in biotechnology, among others [64], whereas many related polysaccharides devoid of gelling ability are investigated as biologically active compounds [91]. Thus, humans possess intestinal enzymes that degrade algal starches to mono-and di-saccharides for transport across the gut lumen, but generally cannot digest dietary fiber (more complex polysaccharides). As mentioned above, algae is a useful source of dietary fiber, for instance, *Codium reediae* (green algae) contains 23.5% of dry weight and 64.0% of dry weight in *Gracilaria* spp. (red algae), values that frequently exceed those for wheat bran, and can be partially fermented by colonic bacteria [64,68]. The undigested materials continue to the large intestine (colon) where microbial co-metabolism ferments substrates such as non-starch polysaccharides, resistant starch, and oligosaccharides to SCFAs (acetate, propionate, butyrate, etc.), and proteins into a wider variety of compounds. The fermentation products can provide nutritional or functional benefits either by being absorbed and transported via the bloodstream or selectively stimulate the growth and/or activity of one or a few beneficial bacteria (probiotics) in the colon, exerting prebiotic effects and influencing digestive outcomes. Several studies have shown the potential of seaweed polysaccharides as prebiotics [99,100].

Abnormal increases in postprandial glycemia are a key risk factor for T2DM and MS [101]. In this sense, some studies have reported the positive influence of fiber on glycemic control in T2DM patients because it reduces postprandial hyperglycemia through delaying glucose absorption in the small intestine [102]. Most studies have shown that soluble fiber partially blocks macronutrient absorption, reduces postprandial glucose response, and beneficially influences certain blood lipids because of their viscous and gel-forming properties [102]. Specifically, soluble non-starch polysaccharides (β-glucans) help to normalize blood glucose and insulin levels, making these polysaccharides a part of dietary plans to treat CVD and T2DM [102]. However, in a prospective cohort study, insoluble cereal dietary fiber and whole grains, instead of soluble dietary fiber, was consistently associated with reduced T2DM risk [83]. In addition, dietary fiber has been related to decreased risk of T2DM by increasing the speed of intestinal transit through mechanical stimulation of the mucosa [103].

The storage polysaccharide of the red algae (Rhodophyta) is known as floridean starch. The floridean starch content depends on the algal species and growth conditions and may reach up to about 35% of dry biomass [91]. Cellulose, a linear (1–4)-linked β-D-glucan, is present in amounts of 2–10% in the cell walls of most red algae, except Nori, which contains insoluble mannans (e.g., in *Pyropia umbilicalis*, *Pyropia tenera*) and xylans (e.g., Palmariales and Nemaliales orders, like *Palmaria palmata*) as insoluble fiber [91,92]. The major polysaccharide components of the red algae are sulfated galactans (carrageenan and agar) that are found in the thallus [104]. Carrageenan is extracted from seaweeds of the order Gigartinales, and from some genera of *Dumontiales*, *Halymeniales*, and *Rhodymeniales* [91]. These polysaccharides are a group of high molecular weight (>100 kDa) consisted three types of carrageenan, named kappa (κ), iota (ι), and lambda (λ) depending on the number of sulfate groups per disaccharide unit, one, two, and three, respectively [105]. Agarans are sulfated galactans in which the α-galactose or 3,6-anhydro-α-galactose units belong to the L-series. Agar, an agaran polysaccharide, is extracted from red seaweeds that include the European-occurring genera *Gelidium* spp. and *Gracilaria* spp. This seaweed extract is widely used in food and in laboratories. It has also been investigated in relation to T2DM [106].

Brown algae (Phaeophyceae) contain sulfated polysaccharides as dietary fiber, e.g., laminarin, mannitol, alginate, and fucoidan, which is not present in any other seaweeds [107]. These compounds constitute the soluble fiber fraction of these algae, except for some alginates, which are insoluble with cellulose. All have their own unique physical and chemical characteristics influenced by species, geographic location, and season, among other factors. Likewise, the cellular quantities and compositions of these polysaccharides vary among species and with seasonal and environmental changes [68]. Phaeophyceae, such as *Laminaria* and *Alaria* species, are characterized by presenting Laminarans (neutral β-glucans), as well known as laminarins. After mannitol, laminarin is the second soluble, indigestible, storage polysaccharide found in algae, thus, laminarin contributes to fiber intake. Laminarin values have being found around 22–49% of the dry matter of the algae [108]. However, its presence depends on many factors, such as the type of algae and the harvest season, as well as environmental conditions such as sea temperature, salinity, ocean currents, depth, and nutrient availability [91]. It is formed by a linear backbone of 20–30 residues of β-(1–3)-linked-D-glucopyranose with some random β-(1–6)-D-glucopyranose side chains [93]. Laminarin can be biochemically modified to enhance its biological activity to be used for cancer therapies, drug/gene delivery, tissue engineering, antioxidant, and anti-inflammatory functions [109].

Alginates, the predominant algal polysaccharides in brown macroalgae (14–40% dry matter of the algae biomass), are obtained from cell walls and intercellular matrices of various algae like *Laminaria* spp. and *Ascophyllum* sp. Further, *A. nodosum* is the brown seaweed that is most exploited for alginate content worldwide. Alginates are a linear hetero-polysaccharides containing β-D-mannuronic acid and α-L-glucuronic acid. Both monomers are linked in a 1–4 configuration and arranged as homogeneous or alternate blocks [9]. The proportion of these blocks determines the physical properties of the alginates. Thus, alginates with high β-D-blocks share have higher viscosity, whereas alginates with high α-L blocks share have better gelling properties. Alginates can be present in alginic acid or salt forms [108]. As with other compounds in algae, the concentration of alginates depends on the species, the part of the thallus, the harvesting season, the depth at which they grow, and the growth stages [10,104].

Fucoidans, a sulfated polysaccharide of brown algae, comprises fucose interconnected by α-(1,3) glycoside bonds, alternating α-(1,3), and α-(1,4) bonds and rarely α-(1,2) bonds. In addition to fucose, it also contains other monosaccharides, including galactose, glucose, mannose, xylose, rhamnose, and uronic acids whose contents vary depending on algal species and season [91]. The average relative molecular weight of fucoidan varies from 7 to 2300 kDa. Fucoidan is a carotenoid that abounds in orders Laminariales and Fucales and, depending on the algae type, it represents approximately 5–10% of algae dry matter, whereas its sulfate content varies between 5 and 38% [108]. Fucoidan extracts from the seaweeds *Fucus vesiculosus* and *U. pinnatifida* (described and specified in the food category with a maximum intake of 250 mg/day) are already accepted as novel food ingredients in the European Union in accordance with its current legislation [110]. Fucoidans are also found in edible species such as *Cladosiphon okamuranus* and *Saccharina japonica* (like *L. japonica*) [68]. Fucoidan is one of the most researched algae molecules in both in vivo and in vitro studies. The highly sulfated nature and molecular weights of marine fucose-containing polysaccharides appear to cause many demonstrated biological activities, including hypoglycemic effects [111].

Polysaccharides known as Ulvans from the green algae display several biological activities of potential interest for therapeutic and nutraceutical applications. These sulfated heteropolymers represent 38–54% of dry algal matter. They are extracted from members of the Ulvales such as *Ulva* and *Enteromorpha* spp. [91].

Among the green microalgal species of *Chlorella* (Chlorophyta), found in both fresh and marine water, are being recognized as particularly rich in polysaccharides [68]. However, many publications on the polysaccharide composition of these algae contain contradictory data [91]. Cyanobacteria (blue-green algae) are especially interesting because they produce different biologically active substances. Some species can produce extracellular polysaccharides with complex structures and unusual content of monosaccharide constituents, including the sulfated polysaccharide Spirulina extracted from *Arthrospira platensis* [91].

The evidence for bioactivity of algal polysaccharides derives largely from in vitro experiments using isolated oligomers/polymers, with less data on testing any whole alga in animal or human trials. As presented later (Section 5 and Section 6), numerous studies indicate the potential for some algal polysaccharides to benefit human health, and in particular with T2DM because of its effect on glucose homeostasis.

### 4.3. Protein and Amino Acid Contents

Algae are considered a viable protein source, with essential amino acid composition requirements [18]. Nevertheless, protein content differs widely among different algae [68].

The quality of proteins in algae stands out for being superior to that of other plant sources, such as wheat or rice. However, its quality is inferior to proteins of animal origin, such as milk or meat [64]. In most analyses of amino acid composition in marine algae, glutamic acid and aspartic acid represent the highest proportions of amino acids [68]. These amino acids occur as protein constituents and as free amino acids or their salts. For humans, glutamate is the major component of the savory, the fifth basic taste called umami from its characterization in kelp. Other amino acids (alanine and glycine) also contribute to distinctive flavors of some marine algae [68]. The non-proteinaceous amino acid taurine is especially abundant in marine red algae (e.g., 1–1.3 g taurine per 100 g dry weight of Nori) and plays a vital role in bile acid and cholesterol excretion contributing to lower plasma cholesterol [112].

Spirulina (*A. platensis*) and various commercial species of the unicellular green alga Chlorella, both with word-wide production, contain up to 70% protein (as dry weight). Among the marine macroalgae, red and green algae (e.g., *Pyropia* spp. (Nori), *P. palmata* (dulse), *Ulva* spp. (sea lettuce)) are often top-choice species for seaweed-based protein sourcing because of their high protein content (47 and 30%, respectively) in contrast to lower levels in most brown algae. However, the protein content of macroalgae decreases in summer, when limitation of nutrients occurs, also affecting the relative proportions of the different amino acids [68].

Plant proteins have been recognized to exert beneficial effects on CVD compared to animal protein [113]. Although evidence in T2DM is reduced, their effect seems primarily related to their effect on lipoproteins and antioxidant properties [114] because of the arginine/lysine ratio. Arginine is the precursor of NO molecule of known antioxidant and antithrombotic properties [20]. Diets high in protein have shown beneficial effects on glucose homeostasis in short-term trials, suggesting its benefits in T2DM prevention [115]. However, results from prospective studies are controversial. Thus, in the Melbourne Collaborative Cohort Study, the relations of total, animal, and plant protein intakes with incidence were tested. This study of 21,523 T2DM participants indicates that higher intakes of total and animal protein were both associated with increased risks of T2DM, whereas higher plant protein intake was associated with lower risk of T2DM [116]. Malik et al. [84] investigated the associations between total, animal, and vegetable protein, showing an incidence of T2DM in 205,802 participants and 15,580 cases of T2DM during a follow-up. The results reflect higher intake of animal protein was associated with an increased risk of T2DM, whereas higher intake of vegetable protein was associated with a modestly reduced risk. The biological mechanisms that support these results could be related to other nutrients/bioactive compounds and variations in the amino acid composition in these foods.

### 4.4. Lipids, Fatty Acids, and the Unsaponifiable Fraction

Total lipid content in marine macrophytes is normally low (between 1.5 and 4% dry matter); thus, it can be accepted that algae is a negligible source of fat and, therefore, it would contribute little to the total energy consumption. Nonetheless, the fat content of algae is higher than those of most consumed vegetables [10,117]. Algal lipids comprise phospholipid, glycolipids, and non-polar glycerolipids, with betaine and some characteristic lipids of a particular genus or species [9,68]. Of these, PUFAs and carotenoid pigments are most noteworthy as functional foods [68]. In addition, algae contains a wide variety of fatty acids and sterols, as well as other unsaponifiable fractions like terpenoids, carotenoids, and tocopherols [10,117]. Regarding fatty acids, ω-3 PUFAs are the most abundant, reaching between 10 and 50% of the total fatty acid content of algae. However, the composition may vary for each species, but also depends on maturation, seawater temperature, and other factors [10].

Lipid membranes contain sterols such as fucosterol (mainly in brown algae) and β-sitosterol (mainly in red algae), which have known health benefits [68]. Embedded in the algal lipid fractions, nutritionally valuable carotenoid pigments such as β-carotene and fucoxanthin are present [68]. Some studies have underlined the importance of algae fatty acids—including ω-3 PUFAs, phytosterols (β-sitosterol), antioxidants (α-tocopherol)— and polyphenols in reducing inflammation and oxidation and determining improvements in the endothelial micro and macro-vascular function. Such effects have preventive roles in T2DM and CVD [118].

Differing from macroalgae, the lipid content of many microalgal species can represent 20 to 50% of the dry matter. However, the available lipid amount depends on the species and cultivation conditions [119].

The amount of fat consumed and its contribution to the total energy has been considered a dietary factor in the development and complications of T2DM. Takahashi et al. [120] studied 417 male T2DM Japanese participants aged 65 years or older and found that a vegetable rich diet, effectively improves metabolism in elderly patients. However, Shah and Garg [81] reported a higher percentage energy intake from fat in patients with T2DM, but non-significant differences among South Asian individuals newly diagnosed with T2DM.

The quality of fat, more important that the quantity, can affect insulin sensitivity and increase the risk of T2DM [121]. In prospective studies, a high intake of SFAs has been shown to increase the risk of T2DM, whereas a high intake of PUFAs reduces it. In addition, SFA replacement by PUFAs lowers blood cholesterol levels and prevents CVD [122]. Forouhi et al. [123] found in a large sample of individuals from eight European countries that high levels of blood α-linolenic acid (ALA, 18:3 ω-3) and linoleic acid (LA 18:2 ω-6) were associated with a lower risk of future T2DM. In contrast, higher levels of other four minor ω-6 fatty acids were associated with higher T2DM risk, whereas marine-origin ω-3 PUFAs were not associated with T2DM risk. However, Brown et al. [124] suggested that increasing ω-3, ω-6, or total PUFAs has little or no effect on the prevention and treatment of T2DM. In addition, experimental studies have shown that diets enriched with ω-3 PUFAs increase insulin sensitivity [125], reduce intrahepatic TG content, and ameliorate steatohepatitis [126]. In the PREDIMED study, the consumption of nuts (rich in unsaturated fatty acids) was found to decrease the incidence of new T2DM cases [127].

The ω-6 PUFAs/ω-3 PUFAs ratio has clinical interest, and some epidemiological studies suggest that a lower consumption of ω-3 PUFAs and a higher ω-6/ω-3 ratio increases the IR through different mechanisms (increasing adipogenesis, decreasing microsomal and peroxisomal fatty acid oxidations, and decreasing the skeletal muscle receptor [128]). Nonetheless, Forouhi et al. [123] did not find any significant association between the total ω-6 to ω-3 ratio and T2DM.

Notably, docosahexaenoic acid (DHA) and eicosapentaenoic acid (EPA) (both abundant in fish oil and in algae) play key roles on metabolic diseases because of their antioxidant and anti-inflammatory properties [126]. As it will be discussed later (Section 5.2), these fatty acids are precursors of eicosanoid and docosanoid molecules with key roles in reducing and resolving inflammatory process [129]. Thus, seaweeds, because of their fatty acid composition, are a food candidate for diet improving in T2DM.

### 4.5. Mineral and Trace Elements

Seaweed algae are a rich source of minerals and essential trace elements (8–40% of algal dry matter), and include calcium, sodium, magnesium, phosphorus, potassium, iron, zinc, selenium, and iodine [24,25,130]. Algae may retain inorganic marine substances as minerals, conditioning their high mineral content. They can also absorb heavy metals, such as arsenic, cadmium, lead, nickel, and copper [64]. Further, their concentration highly depends on different factors such as the seawater quality in which they grow [131].

Discussing the role of each mineral on T2DM would excessively lengthen this work, however, we believe it is one of the least studied aspects and that it requires active investigation. Therefore, clear available evidence about some will be discussed. Many observational studies have demonstrated the link between calcium consumption and a lower risk of T2DM [39]. Related to sodium and potassium, alterations of the Na^+^/K^+^-ATPase pump transport system are believed linked to various complications of T2DM [132]. Zinc deficiency intake might be a risk factor for obesity and T2DM [133]. The role of zinc homeostasis in the pathophysiology of T2DM is known. Among mechanisms involved, the common polymorphism in zinc transporter SLC30A8/ZnT8 may increase susceptibility to T2DM. In addition, altered ZnT8 function may also be involved in the pathogenesis of T2DM [134]. A moderately high dietary zinc intake in 40–55-year-olds was found to reduce T2DM risk by 13% [133]. However, evidence on the association of dietary zinc and T2DM prevention is still limited [135].

Dietary intakes of iron and copper have been associated with higher risk of T2DM. Dietary intakes of total and non-heme iron and copper were positively associated with T2DM risk in 16,160 healthy Japanese, in which they highlight 396 self-reported new cases of T2DM within a 5-year period [136]. Recently, Vaquero et al. [137] tested the hypothesis that patients with obesity and T2DM could have altered iron metabolism. Their results show that in 537 T2DM patients, iron transport and accumulation appear to be altered by T2DM pathophysiological characteristics. These authors found low transferrin saturation in patients with diabesity, which correlated with IR, inflammation, and abdominal adiposity, mainly in women. On the other hand, it has also been observed that a deficient intake of zinc, potassium, calcium, and magnesium could alter glycemic control in individuals with DM2, increasing Hba1C values [138,139].

Selenium and iodine have been related to diabetic pathology and possibly etiology. Antioxidant selenoenzymes and thyroid hormones appear to be involved in glucose tolerance, energy metabolism, insulin signaling, and resistance [140]. In hypothyroidism, whose cause may be iodine deficiency, there is a reduction in glucose-induced insulin secretion by β-cells, and the response of β-cells to glucose or catecholamine is increased in hyperthyroidism due to increased β-cell mass [132]. However, it is important to consider the toxicity of iodine due to excessive consumption [141].

Two extra minerals require special attention, chromium and silicon. Chromium improves insulin sensitivity; inside the cell, chromium binds to apochromodulin and converts to a chromium-chromodulin complex, which binds to the insulin receptor and activates the kinase cascade [142]. The apochromodulin molecule is secreted in response to different stimuli, such as hyperglycemia [143]. However, evidence for the efficacy of chromium supplements has been demonstrated in some T2DM patients but not in healthy individuals [144]. Otherwise, silicon has been suggested to play a key role on some components of metabolism, although many of its properties and mechanisms of action, as well as its recommended intakes are unknown [145]. Our group has worked intensively on the role of silicon in NAFLD, one of the most important manifestations in T2DM [146], and on the role of silicon in postprandial lipemia [147], with hypolipemic and antioxidant effects in aged rats [148], that offer silicon’s enormous potential as a functional ingredient in T2DM [20]. Nonetheless, algae are potentially a source of poisonous minerals, such as arsenic. Our group has reported that algae could contain high amount of arsenic, contributing to reduce the antioxidant capability of algae, by partially inhibiting the antioxidant glutathione system [149].

### 4.6. Vitamins and Related Compounds

Most B vitamins have been linked to T2DM. Algae have been reported to contain vitamins A, B1, B2, B12, C, D, and E, riboflavin, niacin, pantothenic acid, and folic acid [64]. Vitamin C, E, and B12 are present in high concentrations in some algae [10]; however, controversy exist regarding algal Vitamin B12 bioavailability, due to the presence of corrinoids (analog molecules to cobalamin) that do not possess the cobalt ions, and do not present B12 activity in humans [150]. Compared with other phylum, Phaeophyta contain relatively high content of total tocopherol, mostly as α-tocopherol, and Chlorophyta contained a high level of β-carotene [151].

Thiamine (vitamin B1) supplementation may have a positive effect on developing of various diabetic complications [152,153]. The active form of pyridoxine (vitamin B6), pyridoxal-5′-phospate, participates actively in glucose metabolism, acting as a coenzyme for glucose-phosphorylase to use glycogen in the liver and muscle. Although vitamin B6 status is not clearly associated to T2DM development, its deficiency has been suggested to impair some T2DM complications [152,153]. The role of cobalamin (vitamin B12) and folic acid (B9) on T2DM is also controversial. Vitamin B12 and folic acid deficiencies have been associated to hyperhomocysteinemia, a known CVD risk factor; thus, the relationship with T2DM could be suggested. However, in T2DM patients decreased levels of homocysteine has been reported regarding controls [154]. The risk of vitamin B12 deficiency is increasing in vegan individuals and in those individuals with prolonged use of the drug metformin [152,155].

Vitamins with an antioxidant function such as C, A, and E are especially important in the development and complications of T2DM because of their antioxidant activity, positive effect on blood pressure and CVD prevention [10]. Due to the high levels of oxidative stress caused by hyperglycemia in T2DM, increased requirement for vitamin C in patients with T2DM has been suggested [156,157]. However, as in several reports, and despite reactive oxygen species (ROS) overproduction in diabetes, vitamin E, does not give the expected results in preventing cardiovascular outcomes [158,159]. Retinoids have an especially important function as antioxidants, and may be associated with liver lipid metabolism, adipogenesis, and pancreatic β-cell function. Recently, vitamin D and its active form (1α, 25-dihydroxyvitamin D3 (1α, 25-dihydroxyvitamin D3 (1,25 (OH) 2D3)) have been reported to partially normalize insulin release, IR, and systemic inflammation in experimental and epidemiological studies of T2DM [160,161]. However, results of randomized clinical trials on the effect of vitamin D are inconsistent [160]. Vitamin K supplementation, especially K2, seems to originate a T2DM risk reduction by improving insulin sensitivity and glycemic control [162].

### 4.7. Antioxidants: Polyphenols and Related Compounds

Clinical evidence has been reported concerning the effectiveness of polyphenol lignan-rich foods (such as flaxseeds) in reducing insulin, glucose, and serum C-reactive protein levels and improving homeostatic model assessment index of IR (HOMA-IR) in selected patient groups. Further, supporting epidemiological evidence was also reported for total flavonoid intake association with T2DM risk [163]. Flavonoids have also shown a relationship with antidiabetic effects. As in vitro and animal model’s studies demonstrate, they can prevent diabetes and its complications through regulating glucose metabolism, hepatic enzyme activities, and lipid profiles [164,165]. Tannins are a group of relatively high molecular weight phenolic metabolites. Algal lipid fractions contain different types of polyphenols. Brown seaweeds are characterized by polyphenols with special chemical structures and exclusive marine origin, the so-called phlorotannins [71]. These compounds are oligomeric or polymeric phloroglucinol 6 (1,3,5-trihydroxybenzene) derivatives in which phloroglucinol units are connected by aryl-aryl bonds (fucols), ether bonds (phlorethols, hydroxyphlorethols, and fuhalols) or both (fucophlorethols) [166]. In marine macroalgae, catechins and flavanols, among others, are also present as phenolic compounds [118]. Among other bioactive compounds, plant sterols significantly decrease plasma cholesterol and LDL-C levels, even in diabetic rats [167].

Carotenoids have protective effects against T2DM and have shown a role in the treatment of diabetes, via enhancing insulin sensitivity, and its complications such as nephropathy and infectious diseases [168,169]. Among carotenoids, fucoxanthin stands out because of its effects on antidiabetic, anti-obesity, and antioxidant capacity [71]. Fucoxanthin is a characteristic carotenoid of brown algae, including edible species such as *U. pinnatifida* and Hijikia fusiformis (*Sargassum fusiforme*) [151]. In addition, algae belonging to Phaeophyta contain higher amounts of fucosterol and polyphenol than other algae [151]. Polyphenol, fucoxanthin, and fucosterol, with α-tocopherol are especially interesting because of their antioxidant abilities [170,171].

## 5. Effects of Algae Consumption on T2DM Pathophysiology

Main positive effects of algae consumption on T2DM pathophysiology are detailed in the following subsections and summarize in Figure 3.

### 5.1. Glucose Homeostasis

Dietary fiber and polyphenols are among the most widely studied algal compounds for controlling glucose homeostasis. In fact, current studies associate dietary fiber and polyphenols consumption with prevention and management of T2DM. Seaweeds contribute to the daily diet in Korea and Japan, and there is substantial evidence of dietary consumption of different algae and derived food products being associated with low incidence of T2DM in humans [68,172,173,174]. Takahashi et al. [120] studied vegetables that include algae, and observed a significant decrease of HbA1c levels in patients with a daily total vegetable intake of 150 g or more. Furthermore, there was a significant decrease of serum TG levels in patients with a total vegetable intake of 200 g or more. Sometimes, they were linked with improved insulin regulation and sensitivity in human subjects, e.g., *A. nodosum* and *F. vesiculosus* [175], or with a reduction of postprandial glucose concentration and insulin levels, e.g., *U. pinnatifida* [176]. Compound isolates from algae, such as alginate, suppress satiety and to some extent energy intake in most animal and human studies reviewed by Jensen et al. [177]; although it depends on the vehicle applied for alginate supplementation. Furthermore, only one long-term intervention trial found effects on weight loss. Further, Tanemura et al. [178] found that the consumption of mekabu (70 g), the sporophylls of *U. pinnatifida*, consumed with white rice for breakfast reduced postprandial glucose in healthy subjects. These results are consistent with those observed by Yoshinaga et al. [176] which show that Wakame intake combined with 200 g of white rice significantly reduced the postprandial blood glucose response. This specific hypoglycemic effect of these algae could be considered as functional food because could reduce the risk for T2DM in human. However, Kim et al. [179] indicate that *A. nodosum* is more suitable than *F. vesiculosus* as a source of fucoidan to inhibit α-amylase and α-glucosidase activities (enzymes involved in increasing postprandial blood glucose), thus, it is important to investigate the inhibitory enzymatic capacity of each alga. In addition to the preferred algal source, it seems necessary to determine aspect such as time of harvest, extracts from the same species, and doses of treatment because as pointed out in some studies, because they can condition beneficial effects [180,181]. In a specific single blind crossover trial on whole algae, Hall et al. [182] found that adding *A. nodosum* (4%) to bread products decreased the energy intake (16.4%) after a meal in overweight men, whereas no differences were registered in the blood glucose and cholesterol plasma levels.

In addition to human studies, research based on the effects of different algae or their extracts on animal models have been conducted. Thus, some observe how the oral introduction of undegraded λ-κ carrageenan in drinking water (10 mg/L) of 12-week-old mice caused significantly higher peak glucose levels in the blood, leading to a concern that this red alga polysaccharide induced IR and inhibits insulin signaling in mouse liver might contribute to human diabetes. These findings may result from carrageenan-induced inflammation through interaction with toll-like receptor 4 (TLR-4), which is related to inflammation in diabetes [183]. The inflammatory effect may be because of its unusual α-1,3-galactosidic linkage, which is recognized as an immune epitope in human cells, because humans lack the α-1,3-galactosyltransferase enzyme [184]. However, a comprehensive examination of dietary κ-carrageenan effects in rats revealed no effects on blood glucose [68]. Du Preez et al. [185] tested the effect of diets containing carrageenan from the *Sarconema filiforme* in Wistar rats. Animals that consumed algae supplementation modulated GM without changing the Firmicutes to Bacteroidetes ratio. *S. filiforme* improved symptoms of high carbohydrate and high-fat diet-induced MS in rats. The authors highlight the possible mechanisms reduced infiltration of inflammatory cells into organs and prebiotic actions in the gastrointestinal tract. Further, other studies show carrageenan function as prebiotics when supplied as supplements in rat diets [186]. Nonetheless, the potential for algal sulfated galactans to benefit human health remains to be established considering its use is limited for security problems, because it might contribute to increase colonic free sulfate levels [93].

The effects of agar have also been examined and do not seem interesting to prevent diabetes, as Sanaka et al. [187] highlight in their randomized controlled trial. The authors found that agar delayed gastric emptying but did not affect the postprandial glucose response. Nonetheless, Maeda et al. [188] evaluated the efficacy of an agar-supplemented (180 g) diet on obese Japanese patients with impaired glucose tolerance and T2DM, finding that the agar resulted in a marked loss of weight because of the maintenance of a reduced caloric intake and produces a greater reduction in Hb1Ac compared to those who were not supplemented. In addition, total cholesterol was significantly reduced in the agar group compared to the control group, whereas fasting insulin levels were increased.

In another study relating to brown algae, Yang et al. [189] investigated the effect of laminarin on energy homeostasis in mice and conclude that it could be applied to treat obesity and maintain glucose homeostasis. The results showed that chronic oral administration of laminarin (50 mg/mL) treatment significantly decreased high-fat-diet-induced body weight gain and reduced blood glucose levels and glucose tolerance. Laminarin acutely enhanced serum glucagon-like peptide-1 (GLP-1) content inducing GLP-1 secretion in STC-1 cells by triggering the intracellular calcium peak. Further, mRNA expression level of proglucagon and prohormone convertase 1 in the ileum were upregulated. In addition, because of these results, the role of laminarin on T2DM complications had to be considered because it reduces the systolic blood pressure, cholesterol absorption in the gut, and the cholesterol and total serum lipid levels [189]. Continuing with brown algae polysaccharides, alginates with molecular mass over 50 kDa showed a positive prevention effect in T2DM and adiposity [190,191]. Thus, Kimura et al. [190], after studying alginates isolated from *Laminaria angustata Kjellman* var. longissimi Miyabe observed that the sodium alginate significantly prevented hyperglycemia and hyperinsulinemia induced by the glucose tolerance test, but did not affect the small intestinal absorption of glucose. These results are in line with those found in T2DM patients by Torsdottir et al. [192] which showed that sodium alginate also induced significantly lower postprandial rises in blood glucose, serum insulin, and plasma C-peptide. Another study showed calcium alginate markedly inhibited α-glucosidase activity in vitro with the consequent suppression of the postprandial increase of blood glucose [193]. Dietary alginates also provide a sense of satiety and so have been explored as a weight control measure, although there remains uncertainty about its efficacy in this role as highlighted in various clinical trials [194,195,196]. Furthermore, alginates may reduce cholesterol and prevent diabetes, especially by avoiding postprandial peaks of glucose and insulin [19,190,197], and delaying the gastric transit [19]. It should be mentioned that most studies have investigated the effects of polysaccharide extracts rather than consumption of intact seaweeds.

Regarding fucoidans, the highly sulfated nature and molecular weights of marine fucose-containing polysaccharides appear to have many demonstrated biological activities, including hypoglycemic effects [198]. Thus, added to its hypoglycemic effect, several studies have found that fucoidan shows a wide range of positive effects such as antioxidant, anti-inflammatory, and hypolipemic [199,200]. In fact, fucoidan is a popular nutraceutical in Japan and other nations due to their health-enhancing potential [201]. The relevance of fucoidan in preventing diabetes in some animal studies has been highlighted [108,202]. Shan et al. [203] reported the ability of *F. vesiculosus* fucoidan to inhibit α-glucosidase in vitro and to decrease the fasting blood glucose and HbA1c levels of db/db mice. Likewise, Kim et al. [204] demonstrated similar results when administrating *U. pinnatifida* fucoidans [200 mg/mL of different molecular weight forms (5 kDa, 5–30 kDa, and crude)] to the same animal model. Previous studies suggested that different phenolic-rich extracts of *A. nodosum* prefers α-glucosidase over α-amylase inhibition [205,206], although variances in the extraction process could explain studies that found opposite results [207]. Sim et al. [208] studied the effects of fucoidan from *U. pinnatifida* on lipid accumulation, lipolysis, and glucose uptake in 3T3-L1 cells. Sulfated polysaccharide was shown to reduce lipid accumulation and glycerol-3-phosphate dehydrogenase (GPDH) activity in a dose-dependent manner (*p* < 0.01). Expression of PPARγ, a major transcription factor associated with adipocyte differentiation, was also suppressed by fucoidan treatment. These results suggested that fucoidan may have antidiabetic effects by improving insulin-stimulated glucose uptake and inhibiting basal lipolysis in adipocytes without inducing adipogenesis. Yu et al. [209] also demonstrated that fucoidan exhibits an antidiabetic effect by preventing β-cell damage and dysfunction, elevating insulin synthesis by upregulating PDX-1 and GLP1-R through a Sirt-1-dependent manner.

Related to green algae polysaccharides, the reported bioactivities of ulvan extracts in vitro include antioxidant, antihyperlipidemic, and immunoregulatory effects, among others [68]. The antidiabetic effect of ethanol extract of *Ulva prolifera* (2 or 5%) was investigated in high-fat-diet treated mice in drinking water [210]. Some observed that algal supplementation improved glucose tolerance and IR and prevented the increased expression of genes involved in TG synthesis and proinflammatory genes as well as the decreased expression of genes involved in fatty acid oxidation in liver. Although the ingestion of green macroalgae by humans is widespread, the potential health benefits of food supplements of native Ulvans or their chemically-modified derivatives, let alone the direct consumption of the whole algae, are not well understood [211]. In fact, no epidemiological studies have been found linking Ulvans to the prevention or treatment of diabetes.

Regarding the impact of algal polyphenols on T2DM, Xu et al. [12,212] observed an antidiabetic effect of phlorotannins (70% *w*/*w*) from brown alga *Ecklonia kurome* (Laminariaceae) (kurome in Japanese) in KK-Ay mice, the animal model for human T2DM. Algae (0.1%) showed inhibitory in vitro activities on carbohydrate-hydrolyzing enzymes, α-glucosidase, especially the marked inhibition of α-amylase, and decreased postprandial blood glucose levels, tested in vivo. The body weight gain and blood glucose levels in the *E. Kurome* group were lower than the control group. Polyphenols from *E. Kurome* improved glucose tolerance and decreased the fasting blood glucose and insulin levels, fructosamine, and glycoalbumin levels compared with the control group. Likewise, Iwai et al. [213] suggested an antidiabetic and antioxidant effects of polyphenols in a methanolic extract of *Ecklonia stolonifera* (0.2 or 1%), another brown alga, in the same animal model. The active compounds are assumed to be phlorotannins. Moreover, the inhibitory activity of methanolic extract of *E. stolonifera* on α-glucosidase from *Saccharomyces* species was 40-fold stronger than the control, whereas the inhibition of maltase was similar for both. Considering the results, the authors conclude that *E. stolonifera* and its bioactive polyphenols have the potential to be developed as antidiabetic pharmaceuticals and functional foods. Likewise, the antidiabetic properties of *A. nodosum* and *F. vesiculosus* phenolic-rich extract were observed in vivo as the postprandial blood glucose levels and insulin peak decreased 90% and 40%, respectively, on rats under hyperglycemic diets supplemented with 7.5 mg/kg compared to the non-supplemented group [214]. In fact, a human clinical trial showed that the ingestion of 500 mg of a mixture containing *A. nodosum* and *F. vesiculosus* 30 min before the consumption of carbohydrates reduce the insulin incremental area of the curve and an increase in insulin sensitivity, however, not if they only ingested *F. vesiculosus* (500 mg) [175,215].

In relation to carotenoids, alcohol extract of *Sargassum polycystum* (brown seaweed) and water extracts, reduced blood glucose and increased the response to insulin [216]. Fucoxanthin-rich Wakame lipids may ameliorate alterations in lipid metabolism and IR induced by a high-fat-diet [217]. Maeda et al. [217] observed an increased expression of monocyte chemoattractant protein-1 (MCP-1) mRNA expression in mice fed a high-fat-diet. However, MCP-1 mRNA expression was normalized in the high-fat-diet containing fucoxanthin-rich Wakame lipid groups, which promoted mRNA expression of β3-adrenergic receptor (Adrb3) in white adipose tissue and GLUT-4 mRNA in skeletal muscle tissues.

Furthermore, fucosterol has been investigated for development as an antidiabetic agent. Thus, it has been demonstrated that fucosterol stimulates glucose uptake and improves IR by downregulating expression of protein tyrosine phosphatase 1B (PTP1B) and activating the insulin signaling pathway in HepG2 cells [218]. Another mechanism that could explain the antidiabetic potential of fucosterol might be related to the ability of this compound to inhibit rat lens aldose reductase (RLAR), human recombinant aldose reductase (HRAR), protein tyrosine phosphatase 1B (PTP1B), and α-glucosidase. Fucosterol displayed moderate inhibitory activity against RLAR, HRAR, and PTP1B. However, it showed weak or no activity against advanced glycation end product formation and α-glucosidase. In addition, fucosterol showed mixed inhibition against RLAR and HRAR, whereas it noncompetitively inhibited PTP1B. Since fucosterol inhibited aldose reductase (AR), it holds great promise to treat diabetic complications [219].

Unpublished results of our group AFUSAN show that three extracts from edible algae (Sea spaghetti (*Himanthalia elongate*), Wakame, and Nori (*Pyropia umbilicalis*)) present high in vitro GLP-1 secretagogue effects. The acute GLP-1 releasing activity of water and ethanol- and chloroform-extracts of Nori, Wakame, and Sea spaghetti were assessed in STC-1 cells. Clear differences between alga and extracts were observed on the GLP-1 activity. Results showed all Nori extracts, but mainly those of ethanol and chloroform, significantly increased the acute in vitro GLP-1 secretion. Thus, Nori acts as a potent stimulator of GLP-1 secretion in vitro. However, one important thing is the need to avoid any generalization on algae effects as the aqueous spaghetti extract led to the significant decrease of GLP-1 content. In addition, future studies are needed to verify these preliminary results and assess their activity in vivo.

The algae employment as functional ingredients in different food matrices has also been studied [220]. Depending on the study matrix, the glucomannan plus Spirulina consumption has revealed different effects on carbohydrates homeostasis in Zucker fa/fa rats fed an atherogenic diet. The inclusion of these functional ingredients in a fish product made with squid-surimi lowered blood glucose, leptin, and adiponectin levels [221]. In contrast, when glucomannan plus *Spirulina* was included in a meat matrix, no changes were revealed on glycemia [220]. This loss of effect could be due to the higher caloric content of the diets belonging to the study with functional meats, in which the functional ingredient did not show effect on blood glucose.

Several studies in both animals and humans on the potential T2DM health benefits of regular algae consumption and of main representative compounds of algae have been realized and are referenced and detailed in Table 2.

### 5.2. Lipids Metabolism

As previously commented, the atherogenic lipid triad is a major characteristic of T2DM and it is intimately related to IR [33,222]. Thus, major goals for ameliorating the lipemic triad are to reduce central obesity and IR [223], that would help to normalize the TG and HDL-C values; and the LDL characteristics and/or size. Four main interventions have been reported to help to normalize central obesity: (a) getting a negative energy balance by decreasing energy intake and/or increasing energy expenditure; (b) retarding or inhibiting nutrition absorption; (c) modulating metabolic responses; and (d) inducing microbiota abundance and composition [224].

Algae are composed among all of fiber matrix that contributes to the four just commented interventions, as fiber energy contribution is lower than nutrients contributing to decrease the meal energy content. In addition, fiber contributes to decrease nutrient digestion and absorption (overall fat and carbohydrates). Their gel capability formation increases satiety signals and slows down gastric emptying. This last effect contributing to lower TG and glucose postprandial responses, which induces a lower insulin secretion and improves insulin sensitivity helping to decrease IR, the atherogenic triad, the pro-oxidant, and inflammatory status [225]. Finally, fiber can be fermented by GM contributing to assure the instauration and colonization of low inflammatory and obesogenic microbiota that reduce colon and intestinal inflammatory and antioxidant status [224].

According to Maeda [90] algae contain specific polysaccharides used as thickeners and gelling agents for different industrial applications. Those polysaccharides partially suppress absorption of extra fat and cholesterol in the small intestine. The metabolic pathway has been suggested by Sánchez-Muniz [226] as fiber function as bile sequestering agent decreasing the enterohepatic pathway; thus, increasing liver cholesterol synthesis addressed to bile synthesis that contributes to lower cholesterol availability to be included in lipoproteins, contributing to keep low the liver free cholesterol pool and high the apoB LDL receptor.

Unlike what was found on carbohydrates metabolism, the inclusion of functional foods enriched in glucomannan plus Spirulina (2.25% + 0.3%, respectively) did show promising lipid-lowering effects in Zucker fa/fa rats fed an atherogenic diet [227,228]. The formulation of fish or meat matrices with glucomannan plus *Spirulina* reduced hypercholesterolemia, hypertriglyceridemia, and TG-glucose index values of Zucker fa/fa rats fed with these functional products. Thus, this conditioned the lipoprotein profile, presenting VLDL, IDL+LDL, and HDL with lower cholesterol and triglyceride contents compared to control counterparts [227,228]. These data highlight the potential of algae as functional ingredients, demonstrating the maintenance of their lipid-lowering properties even when they are carried in other foods.

Some fiber related compounds as polyphenols have been proposed to increase (up regulates) energy expenditure in brown adipose tissue and reduce the TG formation in this adipose tissue. The mechanism is partially because of the induction of the uncoupling proteins (UCP) that can dissipate energy through uncoupling of oxidative phosphorylation to produce heat instead of ATP. Although the mechanism was firstly related to UCP in brown adipose tissue, adaptive thermogenesis by different UCP has been reported in several tissues (white adipose tissue, skeletal muscle, brain) and proposed as a potential defense against obesity, hyperlipidemia, and diabetes [229]. The UCP activation mechanism seems related to some carotenoids as fucoxanthin [10,90,151], compound that has also been shown to increase of mRNA expression of β-3-adrenergic receptor in white adipose tissue [217]. Reducing the density energy of food and its absorption highly contributes to decrease obesity and IR [33,230]. However other mechanisms have been proposed to contribute decreasing the lipid triad, as to reduce the assembling and secretion of VLDL and to increase the metabolic clearance of these particles. It is well known that high doses of long chain PUFAs ω-3 are beneficial in the treatment of hypertriglyceridemia. In recent years, numerous in vitro and in vivo data have accumulated to suggest that treatment with PUFAs ω-3 could be beneficial in decreasing liver triacylglycerol. PUFAs ω-3 has been proposed to exert a net hypotriglyceridemic effect as they are substrate of microsomal and peroxisomal oxidation contributing to decrease the TG and apoB100 availabilities for VLDL. In addition, PUFAs ω-3 has been found to increase the LPL activity, contributing to increase the VLDL clearance and thus to reduce plasma TG values [231].

In the past few decades, many epidemiological studies have been conducted on the myriad health benefits of PUFAs ω-3 (α-linolenic acid (ALA; 18:3 ω-3), stearidonic acid (SDA; 18:4 ω-3), EPA (20:5 ω-3), docosapentaenoic acid (DPA; 22:5 ω-3), and DHA (22:6 ω-3) [232]. Algae contain variable amount of fat and, thus, of PUFAs ω-3 [10,89,90]. Some algae (*Analipus japonicusas* and *Sargassum thumergii*) are relatively rich in EPA and total PUFAs ω-3 (2.6–3.5 and 9–10 mg/g dry weight, respectively); however, this amount seems insufficient, as 2 g/day PUFAs ω-3 are needed to exert hypotriglyceridemic effects [233]. Similarly, PUFAs ω-3 have been found to exert anti-inflammatory properties through some eicosanoids (e.g., PGE3, resolvin E) and docosanoids (resolving D, maresin, protectin) derived, respectively, from EPA and DHA [129]. However, a large amount of PUFAs ω-3 is needed to decrease the inflammatory effects observed in T2DM. Thus, the efficacy as anti-inflammatory of a normal algae consumption seems limited. Nonetheless, there are nutraceuticals rich in EPA and DHA derived from seaweed and/or seafoods that have demonstrated anti-inflammatory activities [129,234].

### 5.3. Gut Microbiota

The involvement of the microbiome in regulating carbohydrate and lipid metabolism has been discussed in the introduction. The prebiotic effect of algae is one mechanism underlying its antidiabetic properties [235]. There are many in vitro studies and in animal models focusing on the impact of whole algae or isolated component, mostly polysaccharides, consumption on the GM [100,235]. The defined composition of algae allows them to be classified as prebiotics. According to its definition, a prebiotic food serves as “a substrate that is selectively used by host microorganisms conferring a health benefit.” After this definition update, researchers paid attention to the same phytochemicals from seaweed not previously considered as prebiotic. Two interesting reviews have been published that delve into the prebiotic effect of seaweeds, with complete tables summarizing the major results [99,100]. An aspect of great relevance for understanding the complexity of the effects that occur at the colon level is that the influence between the microbiota and certain components of algae such as complex polysaccharides is bidirectional. The fermentation of these components can promote the growth of certain populations of beneficial bacteria, whereas others have been detrimental. Furthermore, how much these components will be fermented and, therefore, the production of bioavailable active metabolites from the algae, depends on the composition of the GM. This makes the prebiotic effects of algae depend on their composition (greatly variable) and on the microbiome of the patient/experimental animal that consumes it [236]. Relating to this question, it is worth highlighting a peculiarity of the fermentation of polysaccharides from algae, which does not affect terrestrial plants. The enzymes responsible for algae polysaccharide degradation (functional carbohydrate active enzymes, or shortly, CAZymes) are usually acquired by horizontal gene transfer, linked to regular consumption of algae. Due to the lack of specific enzymes, non-Asiatic people might not ferment algae when first times consuming them, losing their expected prebiotic effect [100]. Therefore, results about prebiotic properties of algae analyzed in vitro or in healthy animals must be carefully interpreted when discussing their implication relating to their antidiabetic effect. Thus, it is much more accurate to evaluate them in T2DM animal models, as dysbiosis may be presented, seaweed metabolism in this condition will be different, and their effect on microbiota could be changed. It cannot be generalized that prebiotic effects in healthy rodents will be kept in diabetic ones.

Regarding studies about changes in microbiome after seaweed consumption conducted in diabetic models, following the scheme of other sections of this review, results have been organized according to the type of macroalgae analyzed (green, brown, and red algae).

Experiments with green seaweed evaluating microbiota are scarce, even in healthy conditions [237]. Moreover, their polysaccharides are fermented by alfa-L-rhamnosidase in the gastrointestinal tract, and is an infrequent enzyme, which could influence their prebiotic effect. There are two studies, analyzing *Enteromorpha prolifera* consumption (ethanolic extract [238] and flavonoid-rich extract [239]). Lin et al. [238] studied in male mice (specific pathogen-free), the effect of consumption of an ethanolic extract of *E. prolifera* on GM. The results revealed an increase in Bacteroides and *Akkermansia* genus, when Firmicutes and *Turcibacter* decreased abundance. These authors associate the metabolic improvement with the decrease of these bacterial groups, which have traditionally been related with diet caloric absorption and the fat storage by intestinal cells. Likewise, another study conducted with *E. prolifera* flavonoid-rich extract in the same animal model reported an increase in Bacteroidetes phylum, along with a decrease in Firmicutes and *Akkermansia*. In addition, these authors also found an increase in Actinobacteria, Proteobacteria, Alistipes population, Lachnospiraceae group, and *Odoribacter*. These last two groups could affect the release of gut hormones to regulate insulin release, reverse IR, and achieve diabetes control [239]. Thus, the studies evaluating the green algae consumption on GM of T2DM models agree in a greater presence of the Bacteroidetes phylum and a decrease in Firmicutes. However, the results regarding *Akkermansia* genus are contradictory, observing differences depending on the extract consumed. Due to the important and beneficial role of *Akkermansia* in mucous layer and colonic homeostasis, the local consequence of such loss must be further studied. Comparing composition of both extracts, flavonoids are implicated in prebiotic effect, along with Ulvans as the most important compounds in green seaweed. However, Ulvans are poorly or not degraded by fecal bacteria (8.9% of organic matter), indicating that they would be poor sources of SCFAs production in the colon [93]. Besides, algae Ulva, major source of Ulvans, can be rich in free sulfate readily converted to sulfide during fermentation, therefore, consumption of over 20 g/day of whole-dry seaweed may have adverse health effects [68].

Brown algae are the most studied in diabetic conditions in healthy animals. Their most abundant polysaccharides (alginates, laminarin, and fucoidan) have aroused special interest [240]. Alginates and related oligosaccharides have demonstrated prebiotic properties in vivo, by promoting metabolism of the fecal microbiota in humans [236]. *S. fusiforme*, an edible brown alga has shown a remarkable impact on the GM of streptozotocin (STZ)-induced T2DM mice. Relevant results indicate an increase in Bacteroidetes after *S. fusiforme* consumption, especially because of an increase in fermenting species belonging to the *Alloprevotella* genus. The high fucoidans content of this alga and the rise in Bacteroidetes could be responsible for the increase in SCFA-producing species [241]. Shang et al. [78] observed an increase in the abundance of SCFA-producing bacteria, such as *Akkermansia*, *Alloprevotella*, Bacteroides, and Blautia, in mice with T2DM induced with a high-fat diet and fed with fucoidan obtained from *A. nodosum*. These authors justify the metabolic improvement found after consumption of the diet enriched in fucoidans by GM modulation, because some have observed that these compounds improve insulin-mediated glucose uptake in peripheral tissues and increase lipase expression sensitive to hormones, a key enzyme involved in lipolysis [78,240]. *Lessonia trabeculate* is a brown alga of economic importance used mainly to produce alginate. However, Yuan et al. [242] elaborated an extract of this alga with a high content of polyphenolic compounds, to evaluate the effect of its consumption on T2DM mice induced with a diet high-fat diet and STZ. The *L. trabeculate* extract administration markedly increased acetic and butyric acid production compared to the diabetic control. In addition, the bacterial count increased, with a greater presence of the Bacteroidetes group and a reduction of Proteobacteria. The most notable changes in relation to bacterial groups were those found in *Odoribacter* and *Parabacteroides* genus, which reinforces the data obtained from SCFA as they are butyrate-producing bacteria. Interestingly, these bacterial groups have also been associated with better metabolic control in T2DM, as they could have a positive role in regulating IR, inflammation, and intestinal integrity. Thus, this experiment demonstrated the polyphenols from brown algae can be also responsible for prebiotic effects. In fact, phlorotannins presented in brown seaweeds have aroused great expectation [243]. To date, and with animal model studies in mind, brown algae may have greater potential for modulating GM in the diabetic population. Its composition favors the growth of fermenting species that produce SCFAs, which strengthen the epithelial barrier function by promoting epithelial growth, mucus formation, and innate response to pathogens. In addition, they also regulate glucose and lipid metabolism through the activation of FFA receptors/G-protein-coupled receptor (FFAR/GPR) in the liver, adipose tissue, brain, and pancreas; thus, they could contribute to a greater extent to the brown algae antidiabetic effect [244].

Unfortunately, there are no studies evaluating prebiotic effect of red algae on diabetic models. Due to the favorable results obtained in vitro and in healthy animals, studies with red algae and their isolated components (polysaccharides like agarose, carrageenan, and porphyrins, and polyphenols) must be encouraged.

The few studies of algae consumption on dysbiosis of diabetic models indicate promising effects by promoting beneficial species growth and improving glucose metabolism [240]. Specifically, the algae consumption improves the Bacteroidetes/Firmicutes ratio, an indicator of the microbial imbalance widely associated with T2DM development. With brown algae, its polysaccharides rich composition serves as a substrate for the growth of fermenting species, the vast majority belonging to the Bacteroidetes group. In contrast, green algae have a lower polysaccharides content, and more studies are needed to identify the components that show a greater effect on GM. Nevertheless, given the complexity and variety of algae, it would be important to identify the compounds responsible for microbial modulation within the T2DM framework.

### 5.4. Antioxidant Properties

The T2DM multifactorial origin also shows oxidative stress a potential contributor toward this pathology development [159]; this is mainly due to two key factors, diet and the pathophysiology of T2DM itself. The inappropriate eating habits associated with this population, with diets rich in SFA and refined sugars, have proven to be an important source of free radicals and ROS. In addition, this pathology is normally associated with diabetic dyslipidemia and hyperglycemia alterations that induce ROS production and the subsequent redox state disbalance. Therefore, numerous nutritional strategies are aimed at reinforcing the antioxidant balance as a viable treatment against T2DM. As mentioned in the previous (Section 4.7. Antioxidants: Polyphenols and Related Compounds), algae are an excellent source of antioxidant compounds. Any generalization on algal antioxidant compounds must be avoided, as algae composition depends on the species, habitat, and state of maturity, among other factors. Bocanegra et al. reviewed additional information on algae composition [10,89].

Red algae have shown a marked in vitro antioxidant effect [245,246], however, in vivo studies are more limited. Alves et al. [247] studied the effect of the consumption of a lectin isolated from *Bryothamnion seaforthii* on STZ induced diabetic rats, in which they observed an increase in the activity of GPx and plasmatic superoxide dismutase (SOD) in comparison with the control counterpart. Likewise, the consumption of an extract from the alga *Acanthophora spicifera*, rich in flavonoids, increased SOD and catalase (CAT) activities in the liver and kidney, whereas reducing MDA levels in heart, liver, and kidney of diabetic Wistar rats [248]. These polyphenols are widely known for their ability to eliminate free radicals, which would block the oxidative damage of diabetes and would justify the results found.

However, brown algae are the most studied, and numerous studies describe their antioxidant potential, both in vitro and in vivo. Lee et al. [249] identified and isolated Octaphlorethol A, a polyphenol extracted from *Ishige foliacea*. These authors verified its antioxidant potential in vitro on rat insulinoma cell line (RINm5F pancreatic β cells) treated with STZ, where they observed inhibition in lipid peroxidation, ROS intracellular formation, when it increased SOD, CAT, and GPx activities. Likewise, an extract obtained from *Ecklonia cava* has also shown to reduce ROS formation and increase the antioxidant enzyme activities in INS-1 pancreatic β-cells subjected to high glucose damage [250]. The antioxidant status imbalance in the β-cells is a key event in pancreatic dysfunction and the consequent T2DM development, which highlights the protective effect of its consumption. Therefore, these same authors isolated dieckol, a phlorotannin derived from *E. cava* to evaluate its antidiabetic potential in a db/db mouse model. Besides finding numerous metabolic improvements, they observed a marked reduction in liver lipid peroxidation. However, they could not corroborate its effect on antioxidant enzymes found in in vitro studies [251]. Gheda et al. [252] extracted a phlorotannin from the alga *Cystoseira compressa* to verify its antidiabetic effect in diabetic Wistar rats. The results obtained revealed a marked antioxidant effect, reducing MDA levels and increasing total antioxidant capacity, CAT activity, and hepatic GSH levels. Phlorotannins are powerful antioxidants because of their ability to function as ROS scavengers, preventing diabetic-oxidative stress and maintaining adequate GSH levels [253]. Likewise, an extract from another brown alga (*E. stolonifera*) has been shown to markedly reduce plasma lipid peroxidation levels, red blood cells, and liver of diabetic KK-Ay mice [213]. Although this alga is rich in phlorotannins, these authors could not conclude whether the reduction in oxidative damage was due to the antioxidant effect of *E. stolonifera* or is a consequence of its hypoglycemic effect [213].

Different species of green algae such as *Chaetomorpha aerea*, *Enteromorpha intestinalis*, *Chlorodesmis*, *Cladophora rupestris*, and *Ulva lactuca*, have also shown a promising antioxidant effect in in vitro studies [254].A study conducted in diabetic rats induced with STZ and fed with increasing doses of *Spirogyra Neglecta* extract, reported a reduction in lipoperoxides formation, an increase in GSH levels, and the total antioxidant capacity of plasma in a dose-dependent manner [255]. These authors attribute these effects to the high content of polyphenolic compounds present in the extract, which could reduce the GSH consumption and justify its increase compared to control diabetic rats. Another nutritional intervention conducted in 20 subjects with T2DM who consumed pills with equal parts of dry powdered sea mustard (*U. pinnatifida*, Wakame) and sea tangle (*L. japonica*, Konbu) (48 g/day) three times a day for four weeks, significantly reduced TBARS values in erythrocytes, when catalase and GSH-Px activities increased in comparison with the control group [256]. Nonetheless, the antioxidant capability of alga could be negatively affected by the presence of trace elements as arsenic as has been already indicated in this review [149]. Thus, it seems absolutely necessary to have precise information, guaranteed by traceability standards, on clean habitats where algae grow.

One of the most studied microalgae is *Spirulina platensis*, or one of its main components (phycocyanin), whose consumption has been associated with a decrease in ROS and lipoperoxides production, with increases in the levels of the enzymes involved in the glutathione system at the liver level in animals fed an atherogenic diet [257,258]. Furthermore, these same authors also observed a reduction in DNA damage in lymphocytes compared to their corresponding control [257]. Early studies, such as those performed by Gargouri et al. [259,260] in T2DM rats induced with alloxan and fed with *Spirulina* (5%), reported a reduction in liver and kidney lipid peroxidation, accompanied by a significantly reduction in SOD, CAT, and GPx activities in these same tissues, as well as a significant reduction in fasting blood glucose and an increase of glycogen level. In addition, their results indicated Spirulina is efficient in inhibiting hyperglycemia and oxidative stress induced by diabetes. Thus, administration of this alga may help prevent diabetic complications. Furthermore, a randomized study in Korean T2DM patients who consumed 8 g/day of Spirulina reported a reduction in plasma MDA levels, a biomarker of oxidative stress, besides numerous improvements in the lipid profile [261]. In relation to this alga, its antioxidant effects have been associated with its C-phycocyanin content, because it is a powerful ROS scavenger, notably reducing lipid peroxidation [262,263]. Moreover, it also has other antioxidants that can act synergistically and help activate the antioxidant machinery [262].

Algae, besides being food per se, has also often been used as a functional ingredient [85,253,264]. Thus, its antioxidant potential prevails when included in other food. As summarized by Macho-González et al. [20], its application to develop functional meat products has succeeded, because the algae antioxidant properties extend the shelf-life of the meat product and increase the expression and activity of antioxidant enzymes. It is necessary to highlight there are very few studies that delve into the exact mechanism by which algae regulate antioxidant status, although some highlight that it could because of the nuclear factor E2-related factor 2 (Nrf2) activation. This transcription factor regulates gene expression through interactions with the antioxidant response element, which is in the promoter regions of many cell defense genes and has been shown to can be activated by polyphenols [265].

## 6. Conclusions and Future Remarks

The ingredients, fiber, polysaccharides, PUFAs, and polyphenol compounds have been most associated with improvement in T2DM.Mechanisms have been proposed to explain the beneficial effects of the algae on T2DM. However, not all studies fully elucidate the factors involved in beneficial properties of algae dietary intake in managing diabetes.Algae have been postulated as promising antidiabetic agents, effectively reducing carbohydrates digestion and absorption, regulating the α-glucosidase activity.Green and brown algae seem to modulate and restore intestinal dysbiosis associated with T2DM, however, there are no studies evaluating the potential of red algae in relation to this aspect.The large number of bioactive compounds present in algae makes them powerful antioxidant agents, alleviating the oxidative stress associated with T2DM.A variety of algae can be useful for developing functional foods aimed at people with T2DM, reducing the risks of acute and chronic diabetic complications. However, it is important to study the optimization of algae extraction conditions to maximize yields of the active compounds and on the properties of algal compounds useful for the antidiabetic benefits required.Clean habitat where algae grow should be absolutely guaranteed.Further investigations in diabetic individuals are required to elucidate the mechanisms involved in preventing, reducing, and controlling diabetes.Identification of genetic factors affecting dietary response to algal compounds may assist in the development of targeted and potentially more efficient dietary interventions.

## Figures and Tables

**Figure 1 ijms-22-03816-f001:**
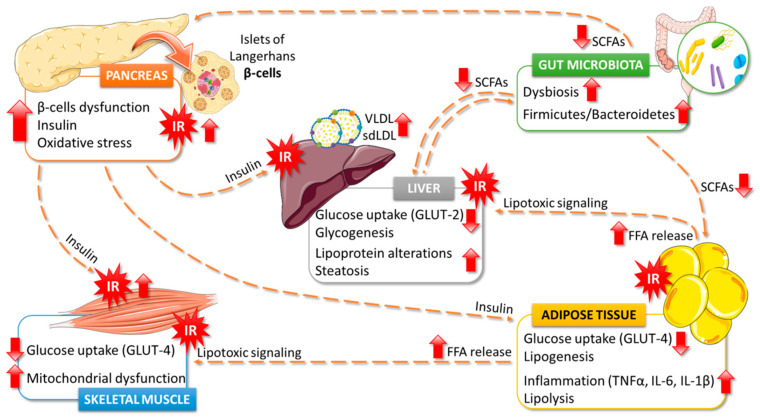
Representative scheme of the Type-2 diabetes mellitus pathophysiology. Main organs affected in T2DM and their interrelation in insulin resistance development. FFA, free fatty acids; GLUT-2, type 2 glucose transporter; GLUT-4, type 4 glucose transporter; IL-1β, interleukin 1 β; IL-6, interleukin 6; IR, insulin resistance; sdLDL, small dense low density lipoproteins; SCFAs, short-chain fatty acids; TNFα, tumor necrosis factor α; VLDL, very low density lipoproteins.

**Figure 2 ijms-22-03816-f002:**
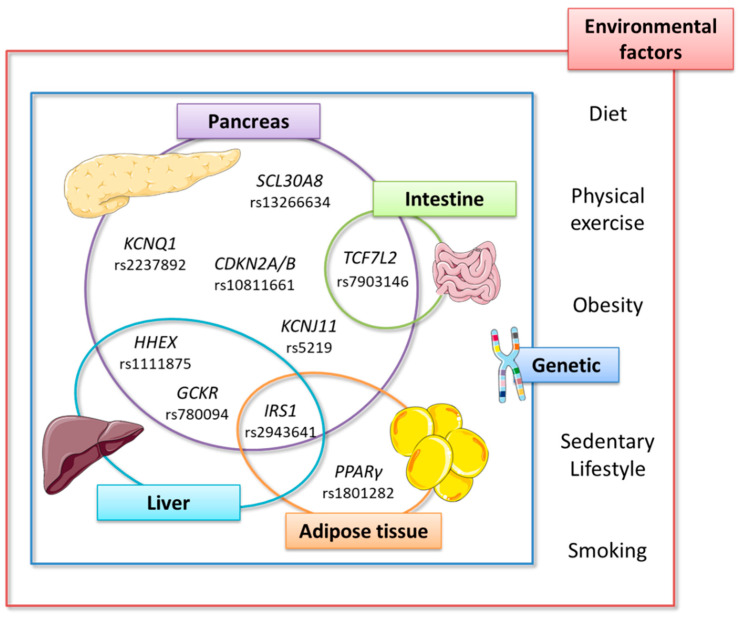
Major environmental and genetics factors related to type-2 diabetes mellitus (T2DM). The Venn diagram summarizes loci significantly associated with T2DM according to the main organ affected (*p* < 5 × 10^−8^). The genetic variants represented are widely influenced by environmental factors, where diet and physical exercise are the main ones.

**Figure 3 ijms-22-03816-f003:**
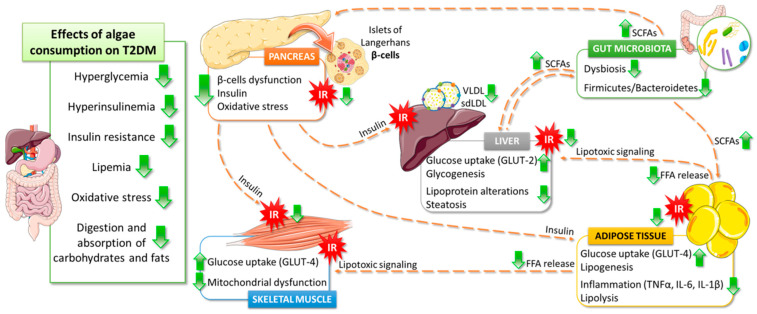
Effects of algae consumption on type 2 Diabetes Mellitus pathophysiology. Schematic representation of how algae consumption is able to modulate the different organs affected in T2DM. FFA, free fatty acids; GLUT-2, type-2 glucose transporter; GLUT-4, type 4 glucose transporter; IL-1β, interleukin 1 β; IL-6, interleukin 6; IR, insulin resistance; sdLDL, small dense low density lipoproteins; SCFAs, short-chain fatty acids; TNFα, tumor necrosis factor α; VLDL, very low density lipoproteins.

**Table 1 ijms-22-03816-t001:** Main characteristics of macroalgal division *.

Division/Common Name	Specie	Pigments	Storage Product	Cell Structural Wall	Intercellular Mucilage
Phaeophyta/Brown algae	*Sargassum polycystum*	Chlorophyll a, c Fucoxanthins β-carotenes Xanthophylls	Laminarans Mannitol	Cellulose, Chitin	Alginic acid/Alginates, Sulfated polysaccharides (Fucoidans)
	*Himanthalia elongata*				
	*Undaria pinnatifida*				
	*Laminaria* spp.				
	*Laminaria japónica*, *L digitata*				
	*Hizikia fusiforme*				
Chlorophyta/Green algae	*Ulva* spp.	Chlorophyll a, b Xanthophylls	Starch	Cellulose, Xylans, Mannans	Sulfated polysaccharides
	*Ulva lactuta*				
	*Ulva pertusa*				
	*Enteromorpha* spp. (*E. compresa*)				
Rhodophyta/Red algae	*Pyropia* spp.	Phycoerythrin Phycocyanin Chlorophyll a β-carotene Xanthophylls	Florideans Starch	Cellulose, Xylans, Mannans	Sulfated polysaccharides (Agar, Carrageenans, Porphyrans)
	*Pyropia tenera*				
	*Pyropia yezoensis*				
	*Chondrus crispu*				
	*Gracilaria verrucosa*				

* Adapted with permission from ref. [10]. Copyright 2021 Copyright MARY ANN LIEBERT, INC.

**Table 2 ijms-22-03816-t002:** Study assay, study characteristic, source, and content and bioactivity.

Study Assay	Study Characteristics	Source and Content	Bioactivity	References
Intervention trial	Diet with large amount of seaweed 417 male Japanese T2DM 65 years or older	Total vegetable intake	↓ HbA1c, Tg, waist circumference	[120]
		>150 g of daily total vegetable	↓ HbA1c	
		>200 g of total vegetable intake	↓ Serum Tg ↓ HbA1c	
		Green vegetable intake	↓ Body mass index, Tg, waist circumference	
Double blind, randomized, placebo-controlled crossover study	23 participants 19–59 year old	Blend of *Ascophyllum nodosum* and *Fucus vesiculosus*	↓ Insulin concentrations ↑ Insulin sensitive	[215]
Randomized Crossover Trial	26 participants	*Undaria pinnatifida* (4 g, dry alga) + Rice (200 g)	↓Postprandial glycemia ↓ Insulin levels	[176]
Randomized crossover study	12 participants	70 g Mekabu(*sporophylls* of *Undaria pinnatifida*)	↓ Postprandial glycemia ↓ Glucose area under the curve	[178]
Randomized controlled trial	12 overweight, healthy males. Aged 40 year	*Ascophyllum nodosum* enriched bread (4%)	↓ Energy intake	[182]
In vitro assay	Brown marine algae from Eastern Canada.	Fucoidan extracted from *Ascophyllum nodosum*	Inhibit α-glucosidade and α-amylase activities	[179]
Mouse model and human hepatic cells	Male C57BL/6J mice 12 weeks	Food additive carrageenan (E-407) Drinking water (10 mg carrageenan/L) HepG2 Cells (1 mg carrageenan/L × 24 h)	↓ Glucose tolerance ↑ Insulin resistance Inhibit insulin signalling	[183]
In vivo assay	Male Wistar rats 16 weeks	Lota-Carrageenans from *Sarconema filiforme* (5%, last 8 weeks)	↓ Body weight ↓ Abdominal and liver fat Improve symptoms of high-carbohydrate, high-fat diet-induced metabolic syndrome.	[185]
Randomized controlled trial	10 healthy male volunteers, studied on three occasions	Agar (2.0 g)	↓ Delay gastric emptying No effect on the postprandial glucose response	[187]
Randomized controlled trial	76 obese patients with type 2 diabetes 12 weeks	Agar (180 g) + Traditional Japanese food	↓ HbA1c, ↓ Visceral fat area, subcutaneous fat area, total body fat, ↓ Insulin area under the curve after oral glucose tolerance test ↓Total cholesterol p.	[188]
In vitro and in vivo assay	Normal C57/BL6 mice 4 weeks	Laminarin (50 mg/mL)	↑ GLP-1 secretion and c-Fos protein expression in STC-1 cells	[189]
In vivo assay	Adult male Wistar rats	Sodium alginate *from Laminaria angustata*. Natural and three water-soluble low-molecular weight	Natural and 50 and 100 kDa molecular weights of alginates: ↓ Glucose tolerance ↓ Cholesterol excretion	[190]
In vitro and vivo assay	Wistar rats	Calcium alginate	Inhibited α-glucosidase activity in vitro Suppression postprandial increase of blood glucose	[193]
Randomized controlled trial	48 overweight or obese participants 10 days	Sodium alginate from *Laminaria digitata*	No effect on gastric motor functions, satiation, appetite, or gut hormones	[195]
Randomized placebo-controlled trial	176 participants 5 weeks	Fiber supplements of alginate + balanced 1200 Kcal diet	↓ Body weight	[196]
In vitro and in vivo assay	IEC-6 cell line Male db/db mice	Fucoidans from eleven species of brown algae	Specially from *Fucus vesiculosus*: ↓ α-glucosidase activity ↓ Fasting blood glucose ↓ HbA1c	[203]
In vitro assay	Human colonic carcinoma Caco-2 cells	Five brown species of alga	Cold water and ethanol extracts of *Ascophyllum nodosum*: ↓ α-glucosidase activity	[206]
In vitro assay	3T3-L1 cells	Fucoidan from *Undaria pinnatifida* Several concentrations	↑ Glucose uptake ↓ Lipolysis ↓ Expression of PPARϒ	[208]
In vivo assay	Diabetic KK-A(y) mice 5 weeks	Polyphenols from *Ecklonia kurome* (0.1%)	↓ α-glucosidase and α-amylase activity ↓ Postprandial blood glucose ↑ Glucose tolerance ↓ Fasting blood glucose ↓ Insulin levels	[212]
In vivo assay	Male diabetic KK-A(y) mice 5 weeks	Polyphenols (Phlorotannins) from *Ecklonia stolonifera* (0–1%)	↓ Increase in plasma glucose ↓ Increase in lipid peroxidation in plasma	[213]
In vivo assay	Male Wistar rats	*Ascophyllum nodosum* and *Fucus vesiculosus* extract (10% polyphenols) (7.5 mg/kg body weight)	↓ α-glucosidase and α-amylase activity ↓ Postprandial blood glucose	[214]
Double-blind, placebo-controlled, randomised croos-overal trial	38 healthy adults (Asian and non-Asian)	Polyphenol-rich *Fucus vesiculosus* extract (500–200 mg)	↑ Risk of insulin resistance among Asian populations	[215]
In vivo	Rat model of type 2 diabetes	*Sargassum polycystum*, alcohol (150 and 300 mg kg (−1) body weight) water extract (150 and 300 mg Kg (−1))	↓ Blood glucose ↓ HbA1c levels ↑ Response to insulin	[216]
In vivo	Obese murine model 10 weeks	Fucoxanthin-rich *Undaria pinnatifida* (Wakame) lipids (carotenoid)	↓ Alterations in lipid metabolism and IR induced by a HF diet	[217]
In vitro assay	Insulin-resistant HepG2 cells	Fucosterol from *Ecklonia stolonifera*	↑ Glucose uptake ↑ Insulin resistance by down regulating expression of PTP1B ↑ Insulin signaling pathway	[218]
In vivo assay	Growing Zucker fa-fa rats 8 weeks	Modified AIM-93 diets containing 30% of freeze-dried skid surimi formulated with glucomannan (30%) or with glucomannan plus Spirulina (30%) without added cholesterol	↓ Hyperglycemia (glucomannan + Spirulina), ↑ Adiponectin/leptin ratio in adipose tissue increase	[221]

HbA1C, glycosylated haemoglobin; Tg, triglycerides; GLP-1, serum glucagon-like peptide-1; PPARϒ, peroxisome proliferator-activated receptor ϒ. Arrows ↑ or ↓ indicates significantly more or less than control, respectively.

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
