# Peer review of "Whole Alga, Algal Extracts, and Compounds as Ingredients of Functional Foods: Composition and Action Mechanism Relationships in the Prevention and Treatment of Type-2 Diabetes Mellitus"

_ijms, 2021, doi:10.3390/ijms22083816_

Round 1

Reviewer 1 Report

The paper "Algal components on type 2 diabetes mellitus: prevention and treatment" presents interesting findings related to the beneficial effects of algae in T2DM. Although the title is quite ambiguous and should be reformulated in order to better reflect the complexity of the paper, the paper itself is quite well written and especially well structured and organized.

The authors make a comprehensive introduction regarding T2DM pathophysiology and algae definition, classification and use. Of high interest is also the very detailed documentation of the coposition of algae, their uses and origin within the types of algae consumed all over the world.  Lastly, the beneficial effects of algae consumption on T2DM pathophysiology is well documented, well structured and on point.

Figures are high quality and well designed. 

However, here are some minor revisions sugested:

line 19 - "applications of algae and its components in T2DM" should be "applications of algae and their components in T2DM

line 21 - "were visited" should be "were interogated/used"

line 28 - "alga" should be "algae"

lines 28-30 - "Special mention is made on precision diets containing alga, 28
considering the potential interaction between gene and algal nutrients and bioactive compounds 29
by using new omics approaches." phrase too general and vague

line 40 - "high-enegry consumption" is a term mostly used for electronics, not for diet, rephrase to "high-calory containing food consumption"

line 95 "vicious cycle"

line 96 - "the situation" should be replaced by a more medical/scientific expression such as "imbalance in glucose metabolism"

line 106 - "helps form TG rich in high
density lipoprotein (HDL) and small dense LDL" is vague and should be rephrased. A clearer role for CETP should be stated.

line 124 - "Peroxisome" should not be written with capital letter

line 281 - "too many countries" not correct

The diversity of algae used commercially for human consumption is interestingly presented.

line 372 - subtytle 4 "Composition of algae-beneficial aspects on Type 2 Diabetes Mellitus" vague and should be reformulated

line 376 - "The role of energy," not clear what the authors mean

line 383 - "alga composition" should be "algae"

line 1271 - "diabetes" should be replaced with "T2DM"

line 1288 - "diabetes" should be replaced with "T2DM"

Author Response

Response to Reviewer 1 Comments

We thank you very much your carefully reading of the manuscript, as well as the encouraging comments and helpful suggestions. All comments and suggestions have been taken into account. We believe that the manuscript has been largely improved by the changes introduced.

Details of change performed are:

Point 1: The paper "Algal components on type 2 diabetes mellitus: prevention and treatment" presents interesting findings related to the beneficial effects of algae in T2DM. Although the title is quite ambiguous and should be reformulated in order to better reflect the complexity of the paper, the paper itself is quite well written and especially well structured and organized.

Response 1: Following your suggestions a new title has been included in the second version of the manuscript in order to better reflect the complexity of the paper. The new title is: Whole alga, algal extracts and compounds as ingredients of functional foods: composition and action mechanism relationships in the prevention and treatment of type 2 Diabetes.

Point 2: line 19 - "applications of algae and its components in T2DM" should be "applications of algae and their components in T2DM.

Response 2:  It has been corrected as suggested.

Point 3: line 21 - "were visited" should be "were interogated/used"

Response 3: It has been corrected as suggested.

Point 4: line 28 - "alga" should be "algae"

Response 4: It has been corrected as suggested.

Point 5: lines 28-30 - "Special mention is made on precision diets containing alga, considering the potential interaction between gene and algal nutrients and bioactive compounds by using new omics approaches." phrase too general and vague

Response 5: As it was suggested, we have included a new sentence: As the responses to therapeutic diets vary dramatically among individuals due to genetic components, it seems a priority to identify major gene polymorphisms affecting potential positive effects of algal compounds on T2DM treatment.

Point 6: line 40 - "high-energy consumption" is a term mostly used for electronics, not for diet, rephrase to "high-calory containing food consumption"

Response 6: Although the term “calory” is amply used, “energy” seems more correct as calory is a caloric unit and responds to old studies firstly performed by Lavoisier working with ice calorimetry. Our body activities demand energy to be performed. In fact, modern food composition tables indicate the content of food both in kcal and kilojules. More, one kcal (one thousand calories) is same to 4,184 kJ. Having a look to prestigious nutrition journal as American Journal Clinical Nutrition, Journal of Nutrition, British Journal of Nutrition, and European Journal of Nutrition, food intake should be expressed as MJ (megajules) and not in kcal. Please, have a look to Garcimartín et al. Silicon-Enriched Restructured Pork Affects the Lipoprotein Profile, VLDL Oxidation, and LDL Receptor Gene Expression in Aged Rats Fed an Atherogenic Diet. Journal of Nutrition 2015, 145(9), 2039-2045. Specifically to the Abstract (page 2039) and Table 1 (page 2040).

Point 7: line 95 "vicious cycle"

Response 7: The word “cycle” has been added.

Point 8: line 96 - "the situation" should be replaced by a more medical/scientific expression such as "imbalance in glucose metabolism"

Response 8:  It has been corrected as suggested

Point 9: line 106 - "helps form TG rich in high density lipoprotein (HDL) and small dense LDL" is vague and should be rephrased. A clearer role for CETP should be stated.

Response 9: The sentence has been enlarged giving extra explanations to clarify the role of CETP in the Diabetes lipoprotein metabolism. The new phrase is: The increase of plasma TG drives the changes of core lipids between TG rich lipoproteins (TRLs) and HDL particles. There is a transfer increase of cholesterol ester and TG between HDL and TRLs by means of CETP, resulting in the triglycerides enrichment of the latter. HDL-TG are good substrates for hepatic lipase and the hydrolysis produces smaller HDL particles. The catabolic rate of the small HDL, is faster than that of normal HDL, resulting in a reduced amount of circulating HDL particles.

Point 10: line 124 - "Peroxisome" should not be written with capital letter.

Response 10: It has been corrected as suggested.

Point 11: line 281 - "too many countries" not correct.

Response 11: Thanks for the appreciation. You are right since the phrase is was not clear. We have changed it to “other countries”.

Point 12: line 372 - subtytle 4 "Composition of algae-beneficial aspects on Type 2 Diabetes Mellitus" vague and should be reformulated.

Response 12: As it was suggested, we have included a new subtitle: Algal composition, structure and beneficial effects on type 2 Diabetes.

Point 13: line 376 - "The role of energy," not clear what the authors mean

Response 13: As it was suggested, the sentence has been modified as follows: The role of dietary energy, macronutrients, micronutrients, and other metabolites bioactive compounds in developing T2DM have been widely studied.

Point 14: line 383 - "alga composition" should be "algae"

Response 14: It has been corrected as suggested.

Point 15: line 1271 - "diabetes" should be replaced with "T2DM"

Response 15: It has been corrected as suggested

Point 16: line 1288 - "diabetes" should be replaced with "T2DM"

Response 16: According to your suggestion the term diabetes has been replaced by T2DM. However, it is indicated that the mistake was in line 1288 but we believe that she/he meant to refer to line 1298.

In addition, we would like to inform you that in subsection 5.4 “Antioxidant properties” has been included a new sentences: Nonetheless, the antioxidant capability of alga could be negatively affected by the presence of trace elements as arsenic as has been already indicated in this review [149]. Thus, it seems absolutely necessary to have precise information, guaranteed by traceability standards, on clean habitats where algae grow.

We have also added two new conclusions:

  • Clean habitat where algae grow should be absolutely guaranteed.
  • Identification of genetic factors affecting dietary response to algal compounds may assist in the development of targeted and potentially more efficient dietary interventions.

Thank you so much again. 

Reviewer 2 Report

This manuscript [ID: ijms-1149852] extensively reviewed prevention and treatment of type 2 diabetes mellitus by algae and its components based on large quantitative references recently. In my opinion, it is suitable to be accepted for publication on International Journal of Molecular Sciences.

Author Response

We acknowledge you have found interesting our study. We also thank very much your comments on the acceptance for publication of our review.